# Your Favourite Park Is Not My Favourite Park: A Participatory Geographic Information System Approach to Improving Urban Green and Blue Spaces—A Case Study in Edinburgh, Scotland

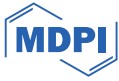

Charlotte Wendelboe-Nelson [1,*] , Yiyun Wang [1], Simon Bell [1,2] , Craig W. McDougall [3] and Catharine Ward Thompson [1]

1   OPENspace Research Centre, Edinburgh School of Architecture and Landscape Architecture, Edinburgh College of Art, University of Edinburgh, Edinburgh EH3 9DF, UK; s.bell@ed.ac.uk (S.B.); c.ward-thompson@ed.ac.uk (C.W.T.)
2   Department of Landscape Architecture, Estonian University of Life Sciences, 51006 Tartu, Estonia
3   Scottish Collaboration for Public Health Research and Policy, School of Health in Social Sciences, University of Edinburgh, Edinburgh EH1 2QL, UK; cmcdoug3@ed.ac.uk
*   Correspondence: cwendelb@ed.ac.uk

**Abstract:** Access to urban green and blue spaces (UGBSs) has been associated with positive effects on health and wellbeing; however, the past decades have seen a decline in quality and user satisfaction with UGBSs. This reflects the mounting challenges that many UK cities face in providing appropriate public facilities, alongside issues such as health inequalities, an ageing population, climate change, and loss of biodiversity. At present, little is known about the preferences of different population subgroups and, specifically, the UGBSs they visit and the spaces they avoid. Using a public participatory geographic information system (PPGIS), the overall aim of the research presented here was to investigate the preferences of different population subgroups in urban areas, and the UGBSs they visit, using Edinburgh, Scotland as a case study. We created a baseline visitor demographic profile for UGBS use, and highlighted how visitors perceive, physically access, use, and engage with UGBSs. The results revealed considerable variation in UGBS preference: one person's favourite UGBS may be one that someone else dislikes and avoids. It is clear that adapting UGBSs to suit local communities should not be a 'one-size-fits-all' approach. The conflicting views and preferences of different groups of respondents point to the importance of developing policies and park management plans that can accommodate a variety of uses and experiential qualities within individual parks. PPGIS approaches, such as those utilised in this study, offer opportunities to address this issue and provide evidence to increase equitable UGBS usage.

**Keywords:** urban green and blue space; community engagement; co-production; public participation; spatial analysis

## 1. Introduction

Many cities in the United Kingdom have seen a significant decline, over the past decade, in the capital and revenue budgets for investment, management, and maintenance of public urban green and blue spaces (UGBSs). This has resulted in a decline both in their condition and in user satisfaction [1]. However, visiting green and blue spaces such as parks, woodlands, rivers, lakes, and coastlines has long been associated with positive effects on health and wellbeing, particularly for those living in relative poverty, for whom high levels of stress and poor mental health are prevalent [2,3]. The importance of urban green and blue spaces for promoting or maintaining health and wellbeing became particularly evident during the 2019 novel coronavirus (COVID-19) pandemic, where the issues surrounding health inequality were also highlighted [4–7]. For example, Hubbard et al. [5] investigated rurality, area deprivation, and access to outside space and green space, and

their associations with mental health during the COVID-19 pandemic. They found that people living in urban areas had poorer mental health compared to those living in rural areas, while increased mental health distress was seen in deprived areas, compared with more affluent ones. In addition, they showed that females and people presenting more severe COVID-19 symptoms, living in deprived areas, may suffer worse mental health distress during such a pandemic crisis [5]. These findings are supported by another study by Hubbard et al. [4], where an association between sociodemographic status and mental health was identified, which was shown to be exacerbated by loneliness, a lack of social support, and thoughts about COVID-19 [4].

Evidence also shows that physical environments have the potential to elicit both pathogenic and salutogenic effects [8], highlighting the importance of the condition and quality of UGBSs in relation to wellbeing. In addition, the literature shows a preference by people for viewing natural over urban scenes [9], and a beneficial effect on psychological wellbeing and cognition from walking in some natural environments [10–13]. Research has also shown that people who live within a five-minute walk of their local green space are 61% more likely to visit once a week or more, compared with those living 5 to 10 min (40%) or 11 to 20 mins' walk away (18%) [14].

Due to constraints on budgets, many UK cities are currently faced with mounting challenges, such as health inequalities, a growing and ageing population, climate change, and a loss of biodiversity, which are increasingly difficult to address [15–17]. Thus, investment in any public facility or service, such as UGBSs, needs to be carefully targeted, with confidence that the investment made will be effective in addressing the needs of the community and wider social challenges.

At present, little is known about the preferences of different population subgroups in urban areas and, in particular, what characterises their preferences for UGBSs. Theories on landscape preference suggest there may be biological, cultural, and personal or idiosyncratic dimensions that underlie people's landscape preferences [18] and it is likely, therefore, that they are complex to understand and diverse in expression. Previous studies have shown that people with different socioeconomic backgrounds tend to visit places likely to be visited by others of similar socio-economic status (SES) to their own, but for some low SES populations, parks are also attractive if visited by users of a higher SES [19,20]. In addition, people of different ages tend to visit different UGBSs [21,22]. How can public authorities adapt UGBSs to better suit the needs of diverse local communities? How can they collect and use public opinion to take action and inform practice at a detailed level? These are some of the important questions that the project 'Thriving Green Spaces' (TGS) set out to investigate.

TGS is a major project led by the City of Edinburgh Council (CEC), with the aim of improving the city's natural environment by producing a 30-year strategy and action plan for protecting and enhancing UGBSs to benefit people both today and in the longer term [23]. The TGS project was funded by a Future Parks (FP) Accelerator project grant, which was successfully bid for in 2019. FP is a joint venture between The Heritage Lottery Fund, the National Trust (NT), and the Department for Levelling Up, Housing and Communities [24], to provide funding to preserve the future of the UK's urban parks and green–blue spaces. The FP Accelerator programme was the first of its kind in the UK, and nine urban areas, covering a population of five million people, were chosen to join this initiative for their ambitious and creative plans to put green spaces at the heart of local communities (Birmingham; Bournemouth, Christchurch and Poole; Bristol; Cambridgeshire and Peterborough; Camden and Islington (in London); Edinburgh; Newcastle; Nottingham; Plymouth). The green–blue spaces across these areas total almost 20,000 hectares and include parks, woodlands, cemeteries, allotments, playing fields, and nature reserves.

The TGS project in Edinburgh focuses on six change categories: community (people, spaces, and operations), ecology, finance, technology, governance, and sharing [23]. The focus of this paper sits within the people and spaces categories. From here on, the abbrevia-

tion 'TGS' refers to two sub-projects: investigating the preferences and visions of different population subgroups in the city, and the green and blue spaces they visit or avoid.

The study design and methods were chosen to elicit information related to UGBS preferences and future visions, e.g., in relation to green space size, location, visitor numbers, management practices, type of vegetation, and biodiversity. The interest in investigating the links between biodiversity and health, in particular, has increased significantly over recent decades [25], and there is an increasing consensus that biodiverse landscapes are linked with various indicators of improved health and wellbeing [26]. On the other hand, there is also evidence suggesting that not all biodiversity–health pathways are positive [27]. Marselle et al. [27] propose four pathway domains by which biodiversity influences human health: reducing harm, restoring capacities, building capacities, and causing harm [27]. By contrast, the biodiversity in urban areas is often threatened by humans, through common management practices such as the maintenance of lawns, pruning of trees and shrubs, use of pesticides and herbicides, and problems of invasive plant species [28].

The overall aim of the study was to investigate the preferences of residents of Edinburgh and their perceptions of the UGBSs they visit. We hypothesized that people are more likely to visit UGBSs if they are located within walking distance from their home; that different age groups visit different types of UGBSs; and that people with different socioeconomic backgrounds visit different UGBSs.

Our study aimed to answer the following research questions:

1. Who visits UGBSs in the city, which UGBSs do they visit and/or avoid, how do they get there, and what activities do they engage in?
2. Are the UGBSs visited and/or avoided distinguishable by the demographic characteristics of the respondents?
3. What are the characteristics of UGBSs that attract or deter people?

## 2. Context, Materials, and Methods

### 2.1. City of Edinburgh

In 2020, Edinburgh's estimated population was 528 thousand people, and this is projected to grow to around 586 thousand by 2043, with the main driver of population growth being immigration [29]. Edinburgh's land area covers 264 km$^2$, with a population density of 2003 residents per square kilometre. It was identified as the greenest city in the UK in 2019, according to a report by First Mile [30], with more than 49% green space, more than 112 parks, and over 750,000 trees. As part of an initiative to support the city to reach its net zero emissions target, the city aims to plant around 250,000 trees over the next decade, and to become a 'one million tree city' by 2030 [31]. See Appendix A for a map of Edinburgh's green spaces and the specific sites included in this study.

The University of Edinburgh, as a partner organisation in the TGS project, was invited to contribute to the project in a number of ways, engaging staff and postgraduate students in research and analysis to support the TGS outcomes. One aspect of this was to undertake a study to answer the research questions outlined above.

### 2.2. Project Design

To ensure public engagement and participation, the TGS project applied the approach of public participatory geographic information systems (PPGIS), specifically a commercially available tool called Maptionnaire® [32], which is a digital community engagement platform built on the principles of participatory action research (PAR), offering a range of tools for public participation. Maptionnaire is based on the Soft Geographic Information System (SoftGIS) methodology developed at Aalto University, Finland (2005). SoftGIS was one of the earliest examples of an advanced online PPGIS approach. PPGIS tools are more widely used today; development has accelerated during the last decade and there is now a wide range of available PPGIS tools [33].

Maptionnaire produces both spatial and non-spatial data. Individual respondents map spatial attributes in the form of points, lines, or polygons (spatial data), in relation

to which a series of questions are answered (non-spatial data) [34]. For the analysis of the Maptionnaire data, we employed the PPGIS data analysis framework proposed by Fagerholm and colleagues [34], consisting of three analytical phases, i.e., *explore, explain, and predict/model*.

Briefly outlined, the *explore* phase focuses on exploring the spatial patterns and on the visual representation of spatial data, employing primarily descriptive and visual analysis [34]. For the *explore* phase, we used the analytical tools built into the Maptionnaire system. The *explain* phase looks more closely at the observations than the *explore* phase, and it is within this phase that spatial and non-spatial data are combined and can be further amalgamated with other geospatial data, while various statistical tests can be conducted, either using the tools contained within Maptionnaire or when exported to quantum geographic information system (QGIS) [35,36]. The *predict/model* phase aims to generalize and make findings transferable across geographical locations and contexts, predicting system models to make inferences [34].

### 2.2.1. Survey Design and Collection of Baseline Data

The intention of the study was, *inter alia*, to undertake data collection to establish people's visiting patterns and preferences for UGBSs in 2020, as a baseline that would then allow the monitoring of progress towards improving UGBSs to better align with community needs and preferences over the next 30 years. The collection of such data was also used to develop indicators, against which change over time could be measured. Maptionnaire was the principal tool used for the collection and reviewing of the baseline data and the indicator data. As the methods needed to be robust and usable for future monitoring, a significant amount of time was allocated to streamlining the survey and adapting it for future use, in ways that eliminated initial complications and would facilitate comparison between baseline data and future data collections.

The structure of the PPGIS questionnaire in Maptionnaire was developed for maximum user clarity and user-friendliness. Appendix B contains detailed information on the structure and content of the survey and some illustrations of the interface. The initial two web pages contained the title, the visual identity of the TGS project, a short project description and aim of the survey, details about who should participate, information regarding anonymity and confidentiality, and contact information for further details. Following the welcome pages, the next page asked if the respondent ever visits parks or open spaces. If the respondent answered 'No', a number of questions were asked to elicit information on the reasons for not visiting. Following this were a number of questions related to demographics and socioeconomic status, such as gender, age, ethnicity, disabilities, employment status, marital status, and household income. The survey then moved on to the map-based part of the questionnaire. The respondents were asked to mark a place close to their home to identify the area where they lived, while protecting their anonymity by avoiding identifying their particular residence. They were also asked to mark up to five locations of the UGBSs they visit most often, and up to five locations of UGBSs they avoid visiting. For each of the UGBSs visited most often, participants were asked a selection of questions related to that individual place, to elicit information regarding preferences and reasoning for visiting. We investigated further what aspects were regarded as satisfactory when visiting UGBSs, and if certain aspects, or the lack thereof, detracted from the visitor experience (Appendix B—survey questions). To elicit this information, participants were asked to rate a number of statements, on a Likert scale, from one (bad) to five (good). For each of the UGBSs the respondents avoided visiting, the participants were asked to note down the reason why, in their own words.

### 2.2.2. Respondent Recruitment and Sampling Strategy

The data collection period ran from July to September 2020, during the COVID period when an online tool was the only option available. The survey could be accessed through the TGS project website. Project partners and stakeholders were encouraged to share an

invitation to respond to the survey, via their own media channels, and all the Edinburgh 'Friends of Parks' groups were contacted directly. 'Friends of Parks' groups are community-level, independent voluntary groups, working to enhance, improve, and promote parks, in partnership with their local authorities. The CEC's Newsbeat channel was used to target council staff, and the CEC and the 'Edinburgh Outdoors' Facebook and Twitter profiles were used to reach a wider audience via social media. Information about the Maptionnaire survey was also circulated to employees at University of Edinburgh.

## 3. Results

To investigate RQ 1, we generated a visitor profile (i.e., who visits UGBSs in the city?) and visiting patterns (i.e., which UGBSs do they visit and/or avoid, how do they get there, and what activities do they engage in?).

### 3.1. Visitor Profiles

In the survey, 531 individual respondents participated. Using Maptionnaire, they mapped 1629 points for UGBS areas they like to visit and 279 points for UGBS areas they avoid visiting. Of the survey respondents, 65% were female and 32% were male (Table 1), 24% were between 55 and 64 years of age, and they were predominantly white (93.44%). The majority were married, in a civil union, or cohabiting (69%), and did not consider themselves living with a disability (90%). Most were also in paid work (65%) or retired (24%) and living in a household with a total annual income of GBP37,000 or above (59%).

**Table 1.** Baseline visitor profile for UGBSs. Characteristics of survey respondents compared to sociodemographic data for Edinburgh, extracted from UK governmental data [a].

| Characteristics | | Survey Respondents | Edinburgh | Test of Difference, *p*-Value [b] |
|---|---|---|---|---|
| Gender *n* = 412 | Female | 65.5 | 48.8 | **0.002** |
| | Male | 31.5 | 51.2 | **0.002** |
| | Non-binary | 0.5 | - | - |
| | Prefer not to say | 2.5 | - | - |
| Age *n* = 366 | 18 to 44 | 26.0 | 54.2 | **0.002** |
| | 45 to 64 | 48.0 | 27.7 | **0.019** |
| | 65 to 74 | 19.0 | 9.8 | **0.000** |
| | Above 75 | 7.0 | 8.3 | 0.733 |
| Ethnicity *n* = 320 | White: Total | 94.0 | 91.7 | 0.682 |
| | Mixed or multiple ethnic groups | 0.5 | 0.9 | 0.775 |
| | Asian, Asian Scottish, or Asian British | 1.5 | 5.5 | 0.142 |
| | African | 0.0 | 1.2 | 0.291 |
| | Caribbean or Black | 0.5 | 0.8 | 0.835 |
| | Another ethnic group | 0.5 | - | - |
| | Prefer not to say | 3.5 | - | - |
| Marital status *n* = 283 | Married, civil union, partner | 69.0 | 38.4 | **0.001** |
| | Single, separated, widowed | 24.0 | 61.6 | **0.000** |
| | Neither of these | 3.0 | - | - |
| | Prefer not to answer | 4.0 | - | - |
| Employment *n* = 293 | In paid work | 65.18 | 59.6 | 0.616 |
| | In education | 3.07 | 15.8 | **0.003** |
| | Unemployed | 1.36 | 3.9 | 0.268 |
| | Permanently sick/disabled | 2.04 | 3.7 | 0.489 |

**Table 1.** *Cont.*

| Characteristics | | Survey Respondents | Edinburgh | Test of Difference, *p*-Value [b] |
|---|---|---|---|---|
| Employment *n* = 293 | Retired | 24.91 | 11.6 | **0.028** |
| | House person/carer | 1.36 | 3.5 | 0.332 |
| | Other | 2.04 | 1.9 | 0.943 |
| Social class [c] *n* = 292 | 1 | 4 | 11.6 | **0.054** |
| | 2 | 12 | 14.1 | 0.681 |
| | 3 | 13 | 14.1 | 0.833 |
| | 4 | 17 | 16.7 | 0.959 |
| | 5 | 54 | 43.5 | 0.288 |
| | | Survey respondents | Scotland | |
| Social class | 1 | 4 | 19.5 | **0.001** |
| | 2 | 12 | 19.5 | 0.181 |
| | 3 | 13 | 19.8 | 0.235 |
| | 4 | 17 | 20.5 | 0.568 |
| | 5 | 54 | 20.7 | **0.000** |

Notes: [a] https://www.edinburghhsc.scot/the-ijb/jsna/populationanddemographics/ (accessed on 12 March 2024); https://www.edinburgh.gov.uk/downloads/file/24244/city-comparisons (accessed on 12 March 2024). [b] *p* values for tests of differences < 0.05 are indicated in bold. [c] Respondents' home location divided according to Scottish quintiles (5th quintile, the 20% most affluent areas; 1st quintile, the 20% most deprived areas).

We compared respondents' individual characteristics to those of the population of Edinburgh as a whole and found that the profile of respondents differed for a number of characteristics. Table 1 shows that the most significant differences between the survey population and the general population of Edinburgh were in relation to gender, age, marital status, employment, and social class.

The respondents were also classified on the basis of the degree of deprivation of their area of residence. The Scottish index of multiple deprivation (SIMD) quantifies the extent to which an area is deprived across the following domains: income, employment, education, health, access to services, crime, and housing [37]. For the maps generated from the data, we used SIMD quintiles, which means that all 6976 data zones (postcodes) were grouped into five bands (quintiles), each containing 20% of the data zones. Quintile 1 contained the 20% most deprived data zones in Scotland, and quintile 5 contained the 20% least deprived data zones in Scotland.

Figure 1 highlights a skewed distribution of respondents' home location: 54% of the survey respondents lived in an area from the top quintile (fifth quintile, the 20% most affluent areas) in Scotland, according to SIMD, and only 4% lived in an area from the bottom quintile (the first quintile, the 20% most deprived areas).

Respondents home location, divided according to SIMD quintile

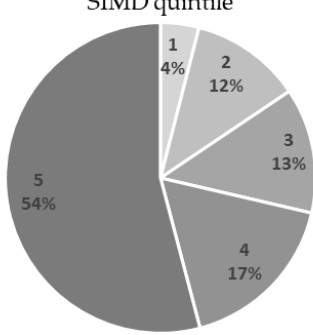

**Figure 1.** Survey respondents' home location, divided according to Scottish Index of Multiple Deprivation (SIMD) quintile, from one (most deprived 20%) to five (least deprived 20%).

### 3.2. Visiting Patterns

Figure 2 shows the UGBS areas the survey population avoided and the areas they visited often. Fewer respondents indicated UGBS areas they avoid (279 points selected), when compared to the number of specified favourite UGBSs (1629 points selected).

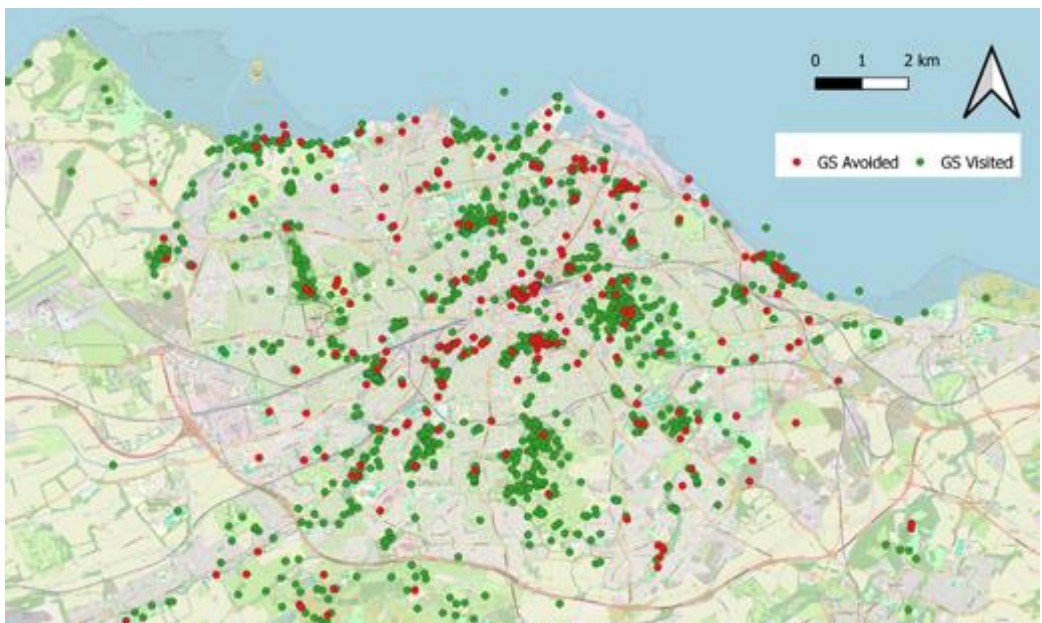

**Figure 2.** UGBSs avoided (279) and visited (1629) by the 531 respondents taking part in the survey.

A count was carried out of the number of times an UGBS was selected as a favourite area to visit, to show the overall preferences for UGBS areas visited and avoided among the survey population. The analysis, represented in Figures 3 and 4, revealed two main UGBS 'hotspots': Holyrood Park, a large natural park in central Edinburgh, with an array of hills, lochs, glens, ridges, basalt cliffs, and patches of gorse, providing a wild landscape within its 260 ha area (Appendix A, green space No. 6); and the Hermitage of Braid and Blackford Hill, a local nature reserve covering an area of 60.3 ha towards the south-western edge of Edinburgh. It is divided into two distinct areas: Blackford Hill, consisting of mainly grass and scrub vegetation; and the Hermitage of Braid, consisting of a narrow woodland dell, with a burn (stream) running through it. The area adjoins a golf course, riding stables, and a riding school (Appendix A No. 23). A number of slightly less-visited hotspots were identified (No. 3: The Braid Hills, No. 8: Princes Street Gardens, No. 7: The Meadows, No. 9: Inverleith Park, No. 12: Corstorphine Hill, No. 21: Calton Hill, No. 10: Harrison Park, No. 11: Saughton Park, No. 22: Victoria Park). The two main hotspots are close to the centre of Edinburgh, with the slightly less visited hotspots circling the city centre.

The UGBS areas respondents avoid (Figure 4) were, to a large extent, in the same places as the areas people like to visit, but with smaller numbers of datapoints. The main UGBS areas respondents avoided were as follows: Princes Street Gardens (Appendix A No. 8), which is over 15 Ha and divided into two parts by an artificial hill, The Mound, that connects Edinburgh's New Town and Old Town and is where the Scottish National Gallery is located; The Meadows (No. 7), a large public park to the south of the city centre, consisting of open grassland crossed by tree-lined paths, a children's playground, a croquet club, tennis courts, and recreational sport pitches; Leith Links (No. 18), which is largely flat expanses of grass bordered by mature trees, containing a children's play area, football pitches, three public bowling greens, and tennis and *pétanque* courts; and Holyrood Park (No. 6).

The variation in visiting patterns between respondents and areas was large, and visitors engaged in an array of different activities of duration, ranging from minutes to

several hours, daily or monthly (Table 2). The majority of respondents visit UGBSs at least once a week (59%), with 8% visiting every day. The most common duration of a visit is for 30 min to 1 h (40%), followed by 1 to 2 h (31%), less than 30 min (15%), and 2 to 4 h (11%). Only a small proportion of visits last between 3 to 5 h (3%), and no respondents said that their visits last more than 5 h. More than half of the respondents walk (64%) to visit UGBSs, 19% cycle, 12% go by car, 5% take the bus, and less than 1% use the tram or another mode of transport (Table 2). The mode of transport chosen when visiting UGBSs indicated that the majority of UGBS visitors live relatively close to the spaces they visit (discussed further in Section 3.3). When visiting UGBS areas, the largest group of respondents chose to visit on their own (42%), 36% visit with their partner or a friend, 17% visit with their family, and 6% visit UGBS areas in a group (Table 2). As the baseline data collection took place in 2020, during different phases of lockdown due to the COVID-19 pandemic, we endeavoured to elicit information related to these new and challenging circumstances. We found that 78% of the respondents were able to continue visiting UGBSs as part of their daily exercise routine during COVID-19 lockdown, with only 22% unable to continue visiting (Table 2). Of the respondents, 67% were able to visit UGBS areas more often compared to pre-lockdown, 19% visited less, and 14% continued to visit as they did pre-lockdown. It is evident that people engage in a wide variety of activities when they visit UGBSs, the main being walking with or without a dog (30%), followed by watching wildlife (12%), meeting friends and socialising (7%), quiet activities such as reading or meditating (7%), and cycling (6%); some people used UGBSs as a through route or for commuting (6%) (Table 2).

**Table 2.** Baseline data highlighting how visitors physically access, use, and engage with UGBSs in Edinburgh.

| | Count | % |
|---|---|---|
| **How often do you visit?** | N = 1423 | |
| Every day | 119 | 8.36 |
| Several times a week | 458 | 32.19 |
| Once a week | 261 | 18.34 |
| Once or twice a month | 350 | 24.60 |
| A few times in the last 6 months | 214 | 15.04 |
| Not in the last 6 months | 21 | 1.48 |
| **Duration of green space visit?** | N = 1019 | |
| Less than 30 min | 154 | 15.11 |
| 30 min to 1 h | 410 | 40.24 |
| 1 up to 2 h | 314 | 30.81 |
| 2 up to 3 h | 111 | 10.89 |
| 3 up to 4 h | 21 | 2.06 |
| 4 up to 5 h | 9 | 0.88 |
| **Mode of transport?** | N = 1213 | |
| Walk | 771 | 63.58 |
| Cycle | 225 | 18.53 |
| By car | 145 | 11.93 |
| Bus | 60 | 4.95 |
| By tram | 3 | 0.25 |
| Other | 9 | 0.76 |

**Table 2.** *Cont.*

|  | Count | % |
| --- | --- | --- |
| Visiting alone or with others? | N = 1017 | |
| On my own | 424 | 41.68 |
| With my partner or friend | 359 | 35.30 |
| With my family | 174 | 17.13 |
| In a group | 60 | 5.89 |
| Ability to continue visiting UGBSs as part of daily exercise routine during COVID-19 lockdown? | N = 1332 | |
| Yes | 1039 | 78.00 |
| No | 293 | 22.00 |
| Activities engaged in when visiting green spaces | N = 988 | |
| Walking without a dog | 230 | 23.30 |
| Watching wildlife | 117 | 11.80 |
| Meeting friends/socializing | 67 | 6.80 |
| Quiet activities (e.g., reading, meditating) | 65 | 6.60 |
| Walking with a dog | 62 | 6.30 |
| Cycling | 60 | 6.10 |
| Through route/commuting | 54 | 5.50 |
| Eating or drinking | 51 | 5.20 |
| Playing with children | 44 | 4.50 |
| Jogging | 41 | 4.20 |
| Art, photography, hobbies | 41 | 4.10 |
| Running | 38 | 3.80 |
| Picnic/barbecue | 28 | 2.80 |
| Sunbathing | 26 | 2.60 |
| Informal games and sports (e.g., frisbee, football, volleyball, etc.) | 12 | 1.20 |
| Visiting an attraction (e.g., a museum in a park or an art installation) | 11 | 1.10 |
| Participating in voluntary activities, e.g., Friends of Parks or other groups | 10 | 1.00 |
| Others | 9 | 0.90 |
| Paddling | 8 | 0.80 |
| Swimming | 6 | 0.60 |
| Adventure sport (e.g., mountain biking, horse riding) | 3 | 0.30 |
| Attending an event (such as a concert or show) | 3 | 0.30 |
| Fishing (including angling and crabbing) | 1 | 0.10 |
| Boating (e.g., yachting, canoeing, kayaking, pedalo/paddle boat, etc.) | 1 | 0.10 |

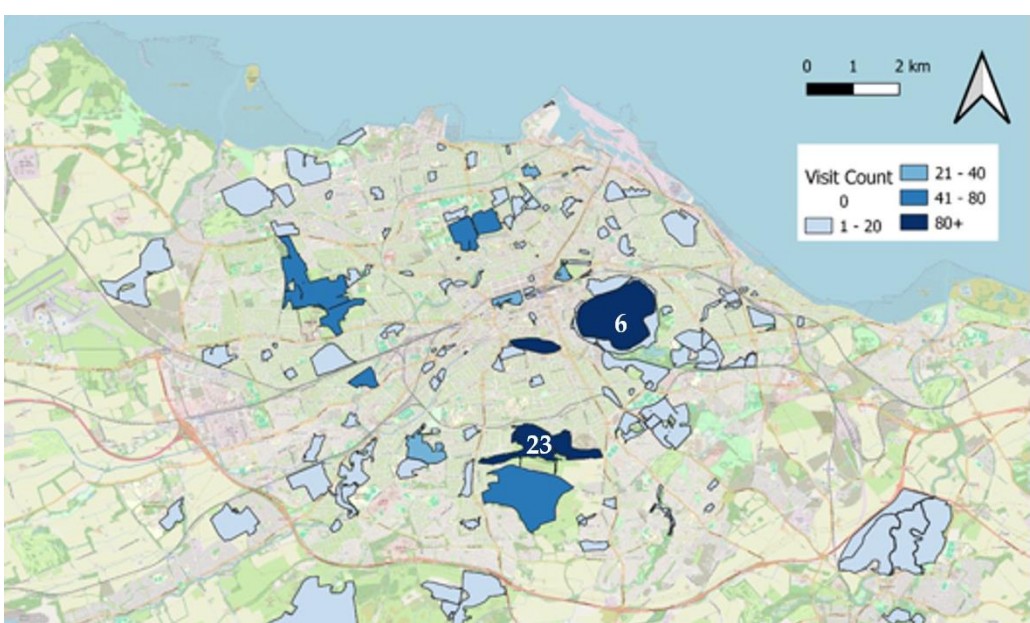

**Figure 3.** The visit count of all the UGBS areas the survey respondents prefer to visit; the darker the blue colour, the more people have chosen the area as a place they like to visit. The map gives an overview of the extended Edinburgh area. Two main UGBS 'hotspots' were identified: Holyrood Park (No. 6); and the Hermitage of Braid and Blackford Hill (No. 23).

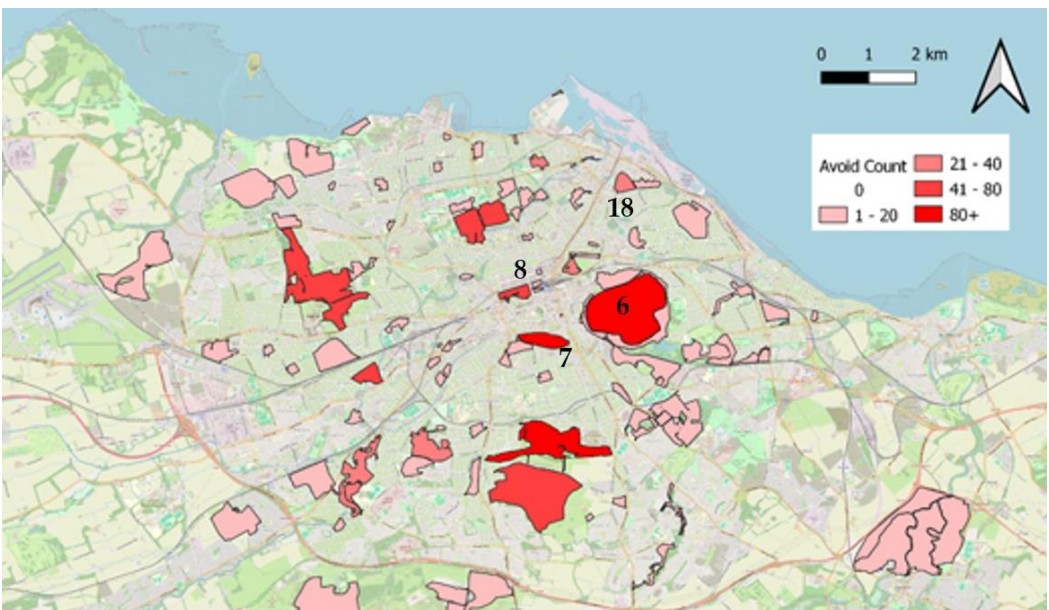

**Figure 4.** The visit count of all the UGBSs the survey respondents avoid visiting. The darker red colours reveal distinct areas that are avoided by the survey population. The main UGBS areas respondents avoided: Princes Street Gardens (No. 8); The Meadows (No. 7); Leith Links (No. 18); and Holyrood Park (No. 6).

### 3.3. Demographic Variability

To address RQ 2, we investigated the plausibility of distinguishing the type of UGBSs visited and/or avoided by demographic information. In Figure 5a, the respondents' home location is shown, colour-coded according to the corresponding data zone from the Scottish index of multiple deprivation (SIMD). According to the maps in Figure 5a,b, it is clear that the majority of respondents live in more affluent areas (Figure 5a); furthermore, according to our data, residents from deprived areas are under-represented when it comes to visiting

and using the UGBSs available in the city. Figure 5b show the UGBS areas respondents liked to visit. Each participant could choose up to five spaces, all of which are coloured according to the SIMD level corresponding to the residence of the person visiting the space. Comparing Figure 5a,b, it appears that individuals from areas of high deprivation tend to visit UGBSs local to them.

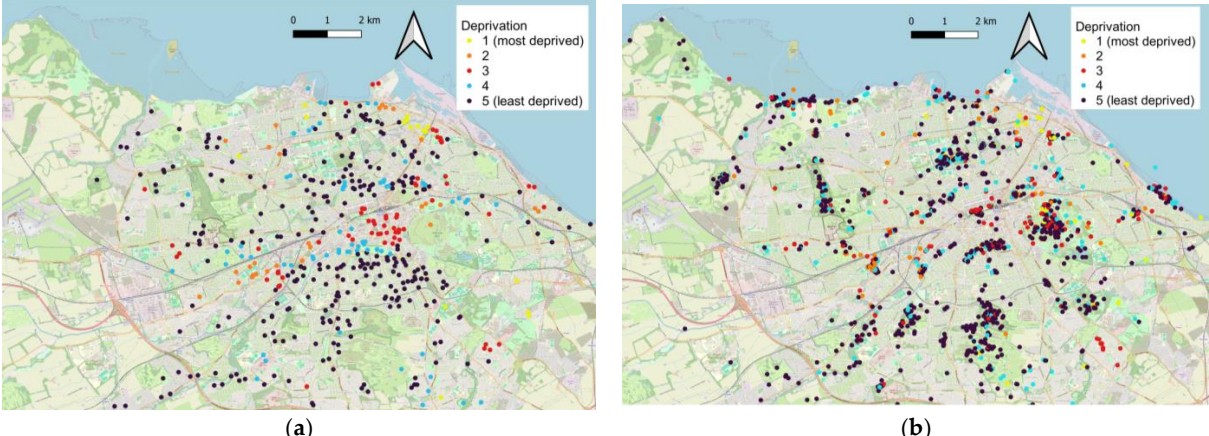

(a)  (b)

**Figure 5.** Maps showing the SIMD level of the area where the respondents lived: Quintile 1 contains the 20% most deprived data zones in Scotland (yellow), and quintile 5 contains the 20% least deprived data zones (black). (**a**) The respondents were asked to mark a place close to their home. (**b**) The five UGBSs that respondents liked to visit. For each participant, their selected pins are coloured according to the SIMD level of their residence.

These findings are supported by the results represented in Figure 6a (distance to UGBSs visited) and b (distance to UGBSs avoided), which shows respondents living in the 40% most affluent areas in Scotland (the top two SIMD quintiles) are more likely to travel further to visit what they perceive to be good-quality UGBSs (with an average distance between 2400 to 2700 m from their home). People living in the 60% most deprived areas travel shorter distances (with an average distance between 1700 to 2000 m from their home) when visiting UGBSs (Figure 6a), and therefore rely more on their local area to provide them with potentially salutogenic environments. In terms of UGBSs the respondents preferred not to visit, individuals living in the 60% most deprived areas explicitly avoided more local areas (with an average distance between 1500 to 2500 m from their home), which could indicate that they do not even consider the possibility of visiting spaces away from their local neighbourhood and therefore do not 'actively' avoid them. In contrast, individuals living in the 40% most affluent areas also considered UGBSs further afield (with an average distance between 3500 to 7500 m from their home) as spaces they avoid visiting (Figure 6b). Combining these findings, it appears that individuals from areas of high deprivation tend to stay more locally than those from areas of lower deprivation.

Focusing on sociodemographic variables can point to places which are more popular among high-income individuals versus low-income individuals (Figure 7), and younger versus older individuals (Figure 8). Figure 7 show groupings based on higher-, medium-, and lower-income survey respondents and the UGBSs they visit. Our data revealed no obvious clusters of usage related to income. The main difference between income groups was that higher-income respondents are more likely to engage in UGBS activities than those on lower incomes.

Figure 8 shows the UGBSs visited, divided into groups based on the ages of the respondent. Again, our data revealed no obvious clusters related to age. What is noteworthy is that the respondents aged 65+ are also venturing as far afield to visit UGBSs as the younger respondents, away from the city centre, visiting more natural UGBSs such as the Regional Park (Pentland Hills, Appendix A No. 20) and the beach/seafront (Appendix A: Nos. 14, 15, 17, and 19).

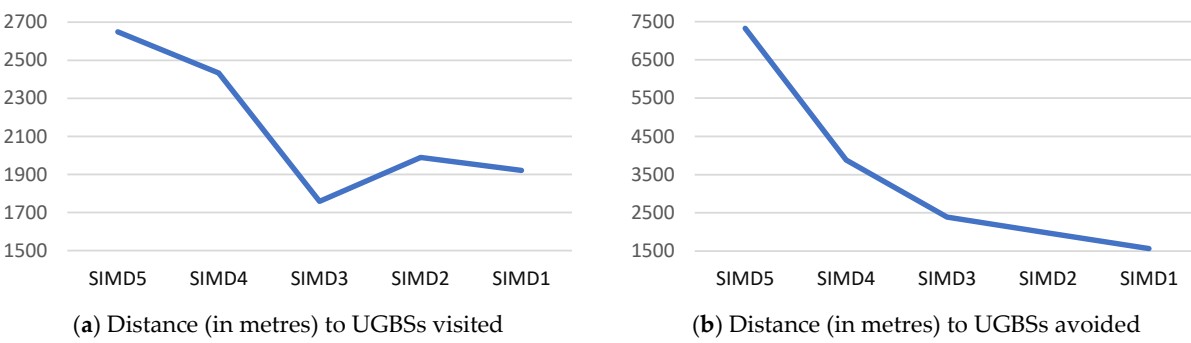

(**a**) Distance (in metres) to UGBSs visited     (**b**) Distance (in metres) to UGBSs avoided

**Figure 6.** (**a**) shows distance from the respondents' home to the UGBSs they like visiting, with the visited spaces grouped according to SIMD category of their residence. (**b**) shows distance from the respondents' home to the UGBSs they avoid visiting, with the avoided spaces grouped according to SIMD category of their residence.

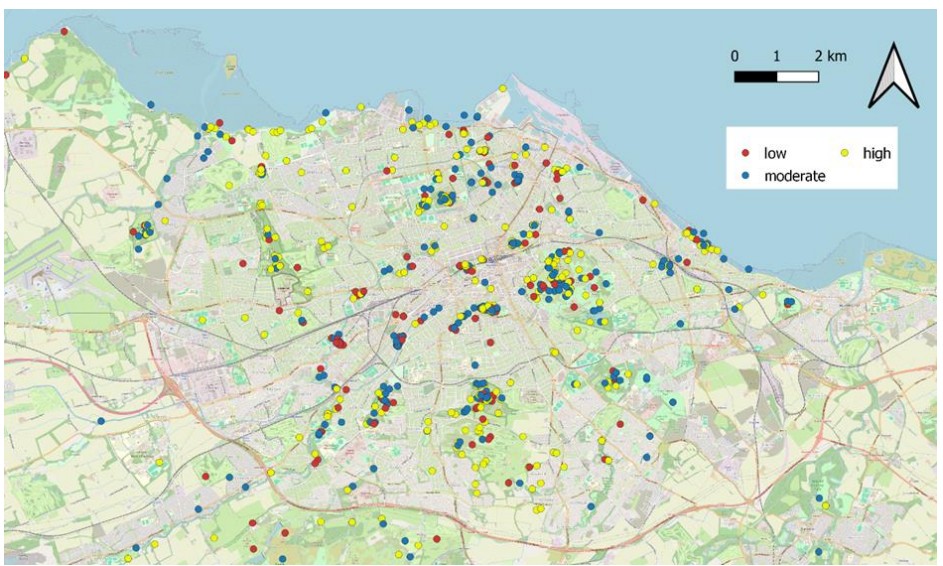

**Figure 7.** The UGBSs visited by respondents according to household income; low (GBP0–GBP26k), moderate (GBP27–GBP45), high (>GBP45k).

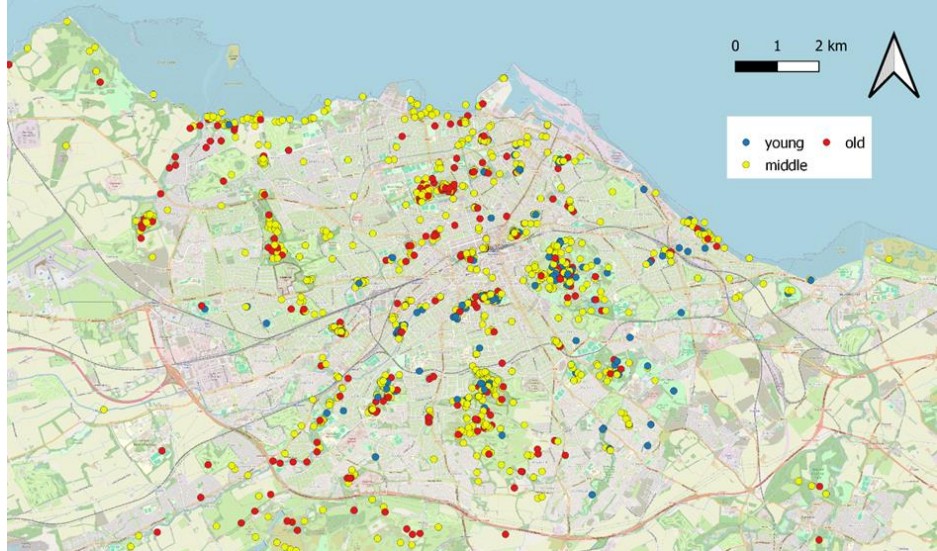

**Figure 8.** The UGBSs visited by respondents according to age; young (16–34), middle (35–64), old (65+).

Figure 9 shows the UGBSs avoided, divided by gender. It is apparent that women are more likely to avoid UGBSs than men, particularly in South and West Edinburgh. The level of avoidance is not correlated with the level of deprivation of the area within which the UGBS is located. The areas typically avoided by both male and female respondents are located towards Leith harbour and the seafront, and near the two main railway stations in the city centre and the railway line. In addition, the areas avoided by females are more evenly spread across the city than for the males, indicating that more areas are perceived as potentially unsafe, or of poor quality, by women, compared with men.

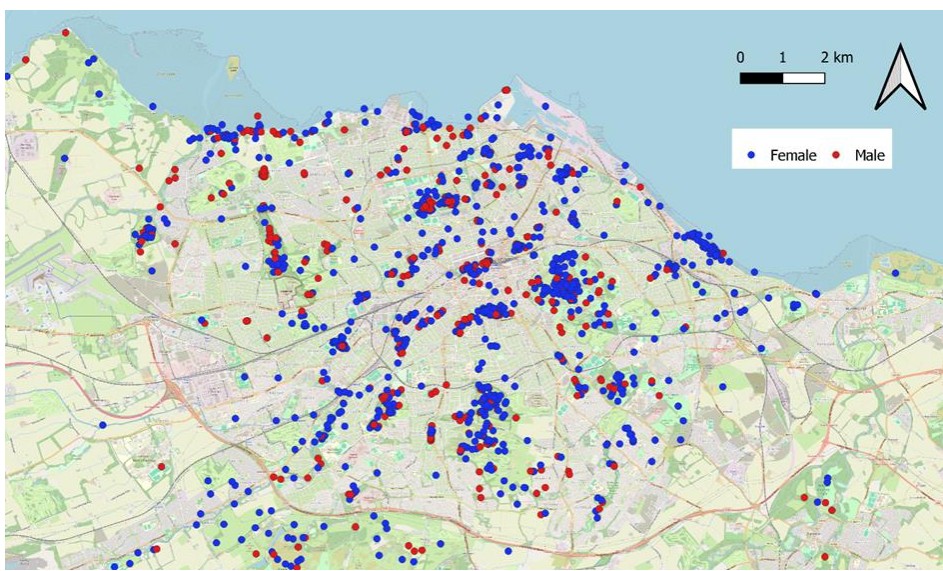

**Figure 9.** The UGBSs avoided, divided by gender.

*3.4. Urban Green and Blue Spaces' Characteristics*

We analysed the open-ended questions from the survey qualitatively, to elicit information regarding perceived barriers and facilitators to UGBS use, and to investigate what specific aspects of the environment limited visitor enjoyment or made the experience more enjoyable (RQ 3).

3.4.1. Barriers to Visiting Urban Green and Blue Spaces

In terms of limiting the experience, the three aspects most often mentioned by respondents as barriers related to: 'dogs/dog owners', 'littering', and 'toilet facilities'. The consensus was that there were too many dog walkers (particularly professional dog walkers) letting their dogs off the lead and not picking up faeces after the dogs (*"Uncontrolled dogs"*, *"Dog owners not clearing dog faeces"*, *"Too many dog walkers, particularly the professional ones"*, *"Out of control dogs are a real issue"*, *"Professional dog walkers with dogs off leads"*). In general, people thought that there was a problem with dog faeces or dog waste bags not being cleaned up (*"The bin is often overflowing with waste, particularly dog waste bags" "Lots of dog poo"*, *"Dogs off lead jumping up, dog owners irresponsible, dog poo bags in prominent places, dogs chasing wildlife"*). However, there were also barriers and limiting factors to dog walkers' experience of enjoyment while visiting UGBSs: *"Also rangers have ruined it for dog walkers, saying to keep dogs on leads because of nesting birds"*, *"Cyclists way too fast and a danger to dog walking…."*.

A second major barrier was 'litter/waste/rubbish', with respondents believing there was too much rubbish lying around, overflowing, and too few bins available, and the bins that were available were not emptied often enough: *"People leaving rubbish …….more or bigger bins are also needed"*, *"People leaving litter"*, *"Overflowing bins"*, *"Trash removal-especially the last few weeks with huge crowds and overflowing litter bins"*, *"Not enough rubbish bins"*, *"Litter bins aren't collected often enough"*.

The third major barrier to people enjoying their UGBS experiences was the lack of toilet facilities: "No toilet facilities which is difficult with children", "No toilet facilities limit how long we can spend there", "Toilet facilities are too far away and it is difficult to park".

### 3.4.2. Facilitators for Visiting Urban Green and Blue Spaces

There was broad agreement among the respondents on which characteristics and aspects of UGBS environments act as facilitators and encourage individuals to visit more often. There were two major facilitators: open space/open views (prospect) and defined areas (e.g., area for children, wild nature for walking, nature-watching, little nature garden, formal garden, orchard, herb garden, dog-free zone, designed for walking, sports facilities, etc.).

### 3.4.3. Urban Green and Blue Spaces Visitor Satisfaction

The survey respondents were asked how much they agreed or disagreed with three statements relating to the UGBS areas they visited: "*I was satisfied with the visit*", "*I felt relaxed*", and "*I felt energised*". For all three statements, the majority of respondents strongly agreed' (68%, 63%, and 59%, respectively), or 'slightly agreed' (23%, 24%, and 22%, respectively). In summary, the participating UGBS visitors in Edinburgh are highly satisfied overall with the UGBS areas they visit. The respondents were also asked a series of questions related to the individual places they visited, to elicit information regarding preferences and reasons for visiting (Figure 10).

In general, visitors were mostly satisfied with the transport links to the UGBSs they visited (Likert scale: 5 + 4 = 52%; 3 = 28%; 2 + 1 = 20%) (Figure 10). They also found the walking conditions to be very accessible, both to and within the site, in terms of quality of pavements, slopes, surfaces, street crossings, etc. (Likert scale: 5 + 4 = 74%; 3 = 18.5%; 2 + 1 = 7.5%). In addition, respondents found the visual quality of the UGBSs they visited to be excellent (Likert scale: 5 + 4 = 81%; 3 = 14%; 2 + 1 = 5%) and were generally happy with the quantity and quality of the trees, shrubs, and flowers (Likert scale: 5 + 4 = 70%; 3 = 18%; 2 + 1 = 12%); they also felt connected or very connected with nature when visiting these spaces (Likert scale: 5 + 4 = 76%; 3 = 14%; 2 + 1 = 5%). They were also largely satisfied with the maintenance of the site (litter collection, condition of equipment, mowing of grass, etc.) (Likert scale: 5 + 4 = 59%; 3 = 24%; 2 + 1 = 16%), and the safety and security (sense of being threatened, poorly lit with dark places in winter evenings, signs of anti-social behaviour) (Likert scale: 5 + 4 = 59%; 3 = 24%; 2 + 1 = 17%) (Figure 10).

Lower satisfaction was seen for the information available online (Likert scale: 5 + 4 = 19%; 3 = 20%; 2 + 1 = 62%); the information/translation of information on site (Likert scale: 5 + 4 = 32%; 3 = 25%; 2 + 1 = 42%); and the facilities (e.g., seating, play equipment, exercise equipment, toilets) (Likert scale: 5 + 4 = 36%; 3 = 35%; 2 + 1 = 27%).

We investigated further what aspects were regarded as satisfactory when visiting UGBSs, and if certain aspects, or the lack thereof, detracted from the visitor experience (Figure 11). The majority of respondents reported 'No' or 'Don't know' when asked if special signs were provided for people with disabilities (30% and 68%, respectively) and if information boards were translated into different languages (38% and 68%, respectively). This indicates that the visibility of this type of information could be improved, or that individuals not needing this type of information will consequently not look for it. Of the respondents, 26% reported that there were toilets in or near the UGBS; 56% reported 'No' to that question, while 32% reported that the provision of toilets influenced their decision to visit a particular UGBS. When asked if any cafes were located in or near the visited space, 50% answered 'Yes' (No = 43%, Don't know = 7%); 24% answered that the provision of cafes does influence their decision to visit an UGBS.

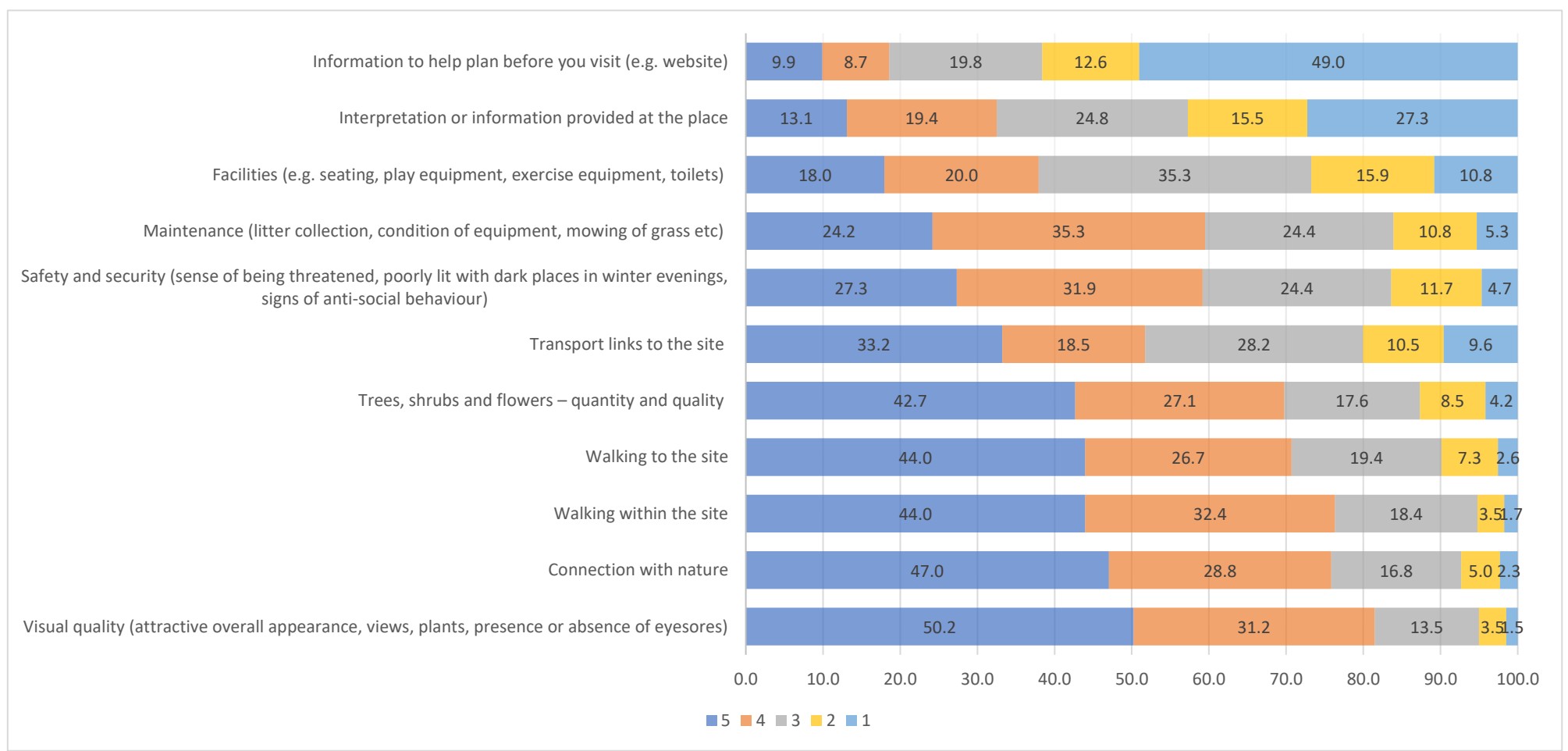

**Figure 10.** Respondents' preferences and reasons for visiting a green/blue space (Likert scale, wherein five is most positive and one most negative).

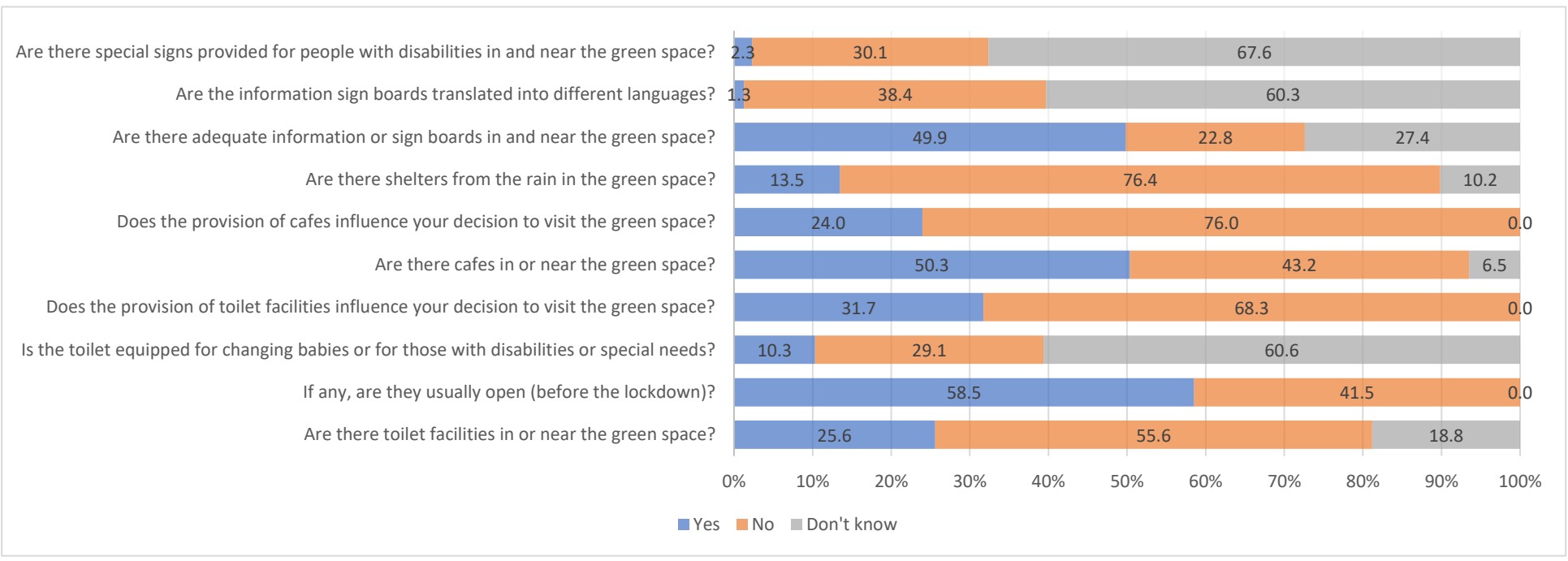

**Figure 11.** Responses to questions about facilities and information provided in green/blue spaces visited.

## 4. Discussion

The overall aim of the research presented here was to investigate the preferences of Edinburgh residents and the UGBSs they visit. These data are of interest primarily because they provide a citywide baseline for the use of UGBSs, underpinning the development of a profile for UGBS access, use, and engagement. The resulting visitor profile shows which respondent groups are visiting UGBSs most frequently and highlights groups that currently do not appear to be fully benefitting from the use of UGBSs. From the visitor patterns showing how visitors physically access, use, and engage with UGBSs in Edinburgh, it was clear that respondents have a range of different preferences for their uses of UGBSs, and that no UGBS area is likely to accommodate all these different preferences within a relatively small geographical area.

### 4.1. Visitor Patterns of Urban Green and Blue Spaces Use and Demographic Variables

We found that the variation in UGBS visiting patterns between individuals and areas was large, and respondents engaged in an array of different activities for any length of time ranging from minutes to several hours, either daily or monthly.

Although some population groups were underrepresented in the survey data, based on our survey, we found that patterns of use are not clearly reflected in the demographic characteristics of respondents. Initially, we hypothesized that different age groups might visit different types of UGBSs and people with different socioeconomic backgrounds might visit different UGBSs. However, our data revealed no obvious clusters of usage related to age or to income. The main difference between income groups was that higher-income individuals are more likely to engage in UGBS activities than lower-income individuals. In addition, it is noteworthy that respondents aged 65+ are also venturing further afield than some younger age groups, away from the city centre, visiting more natural UGBSs such as the Regional Park and the beach/seafront. The literature suggests that, in comparison with young adults, older people may be more concerned with landscape characteristics like legibility, accessibility, safety, and the quality of trails and show less interest in park social participation or vigorous physical activity [38]. Nonetheless, a recent review [38] also found that older people show quite varied opinions on landscape characteristics, some even contradictory to others. Our findings reflect this variation in preferences which, in our sample at least, may not be significantly different across age groups, while acknowledging that older people, especially those of a higher SES, may have the time and resources to travel some distance to find a UGBS that suits their particular preferences.

The lack of engagement (in terms of UGBS visit numbers) with UGBSs in lower-income communities may be explained in part by lower numbers of respondents from deprived areas. However, it may also be partly explained by findings from previous research, which have indicated that the provision and quality of UGBSs vary across the socioeconomic scale, with deprived and/or black and minority ethnic groups having poorer access to good-quality UGBSs [39–42]. In addition, Floyd and colleagues' work in the USA [42] found that specific physical park features were as important as income or ethnic group, in relation to levels of physical activity in parks. Poor provision and quality of UGBSs in an area of high deprivation might therefore cause a cumulative negative effect, making it particularly difficult to engage these communities in UGBS activities.

### 4.2. Urban Green and Blue Spaces Characteristics—Barriers and Facilitators

The data demonstrate differences in respondents' use of UGBSs and whether they see an area as good, or a place to avoid. In particular, the choices related to UGBSs in deprived versus affluent areas are worth further exploration. Our findings suggest that, in order to increase the use of UGBSs in more deprived areas, not only would the aesthetics and usability of the UGBS areas have to be considered in light of the needs and preferences of the local communities, but targeted action would also be necessary to engage local residents, encouraging them to actively participate in their local community, place making, and development of local UGBS areas.

Our findings support the initial hypothesis that people are more likely to visit UGBSs if they are located within walking distance from their home, as 83% of respondents in this study either walk or cycle to the UGBS they visit (Table 1). Of particular interest is the finding showing that people living in more disadvantaged areas visit green spaces closer to their home, but they also tend to avoid spaces that are closer to their home, compared with people from more affluent areas. This indicates that the proximity and quality of UGBSs is a bigger barrier in some areas than others. In addition, we found that women are more likely than men to avoid UGBSs. It is evident that the UGBSs avoided by females are evenly spread across the city, indicating that more areas are perceived as potentially unsafe, or of poor quality, by female UGBS users, compared to the perception of male UGBS users. There is therefore a need for gender-focused UGBS research to minimise gender-related barriers. Our findings highlight the importance of useable and safe UGBSs that are accessible to everyone, but particularly for those living in disadvantaged areas, who visit, or avoid, UGBSs closer to home and are, therefore, likely to benefit more from local community interventions.

The data clearly illustrate that one person's favourite UGBS is not necessarily the same as someone else's, and the same goes for the UGBSs people avoid. The three aspects most often mentioned by respondents as barriers were related to 'dogs/dog owners', 'toilet facilities', and 'littering'. However, the first two barriers were also mentioned by some in relation to facilitators; e.g., for some, being able to take their dog off the lead was a facilitator, whereas, for others, dogs off the lead were seen as a barrier. Some people want access to toilet facilities and cafés, the provision of which influences their decision to visit the space. For others, this is not important, or might even be perceived as a barrier, as it would attract too many visitors, in particular families or groups with children.

There was broad agreement among the respondents regarding two major facilitators: open space/open views (prospect) and defined areas (e.g., area for children, wild nature for walking, nature-watching, little nature garden, formal garden, orchard, herb garden, dog-free zone, designed for walking, sports facilities, etc.). However, again, what for some was seen as a facilitator to visiting UGBSs, was by others seen as a barrier (e.g., wild nature for some is inaccessible; prospect might limit the abundance of biodiversity and feeling of wild, undisturbed nature; loud children and dogs off the lead might scare away wildlife, etc.). Such findings reflect theories on evolutionary bases for landscape preference [18], as well as a recognition of socio-cultural and personal influences on preference and behaviour [43,44].

There was broad agreement that the quality and quantity of greenery, the feeling of connection with nature, and a good overall visual quality, are all aspects that enhance the UGBS experience and act as facilitators for visiting (Figure 10). However, it is not possible from our data to ascertain what individual respondents mean by a good or bad quality of greenery, how they define 'a connection with nature', or how they rate an overall good visual quality. Therefore, in accordance with previous research [45–47], we conclude that adapting UGBSs to suit local communities should not be a 'one-size-fits-all' approach. A solution to accommodate the variation in individuals' preferences for UGBSs could be to section parks into smaller defined areas with different types of vegetation, an array of functions, that targeted towards diverse groups of visitors. Good examples of this are Saughton Park (Figure 12) and Inverleith Park (Figure 13), where there are areas targeted towards families with children, teenagers and young adults, individuals engaging in quiet activities, individuals exercising, and individuals socialising. Saughton Park (Figure 12) recently underwent major regeneration with the support of the Heritage Lottery Fund (HLF) 'Parks for People' and the council capital programme. Wide-ranging public engagement and research was carried out, to inform the development of the Master Plan proposals [48].

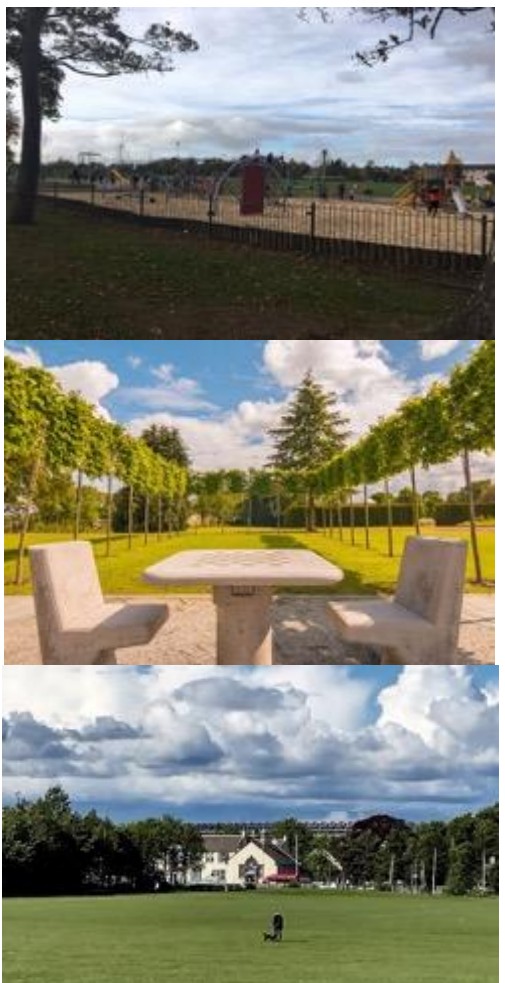
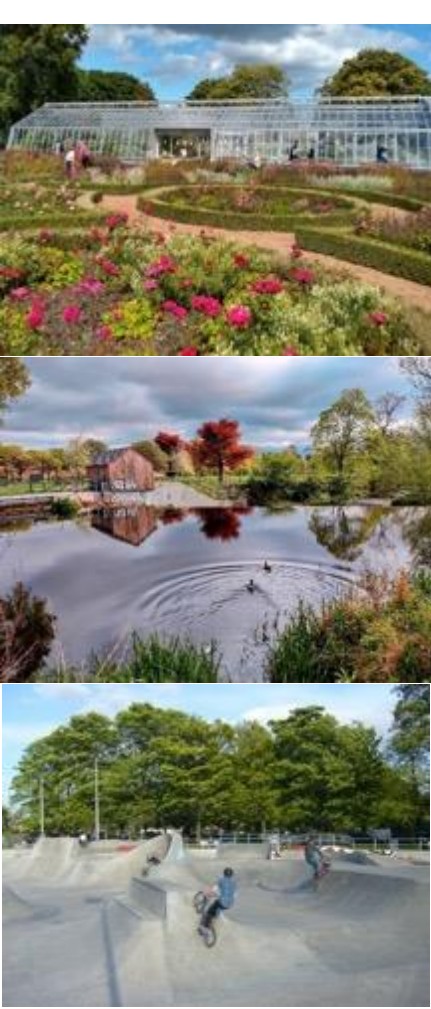

**Figure 12.** Images of Saughton Park, illustrating the use of zoning to accommodate the variation in individuals' preferences for UGBSs (source: the authors).

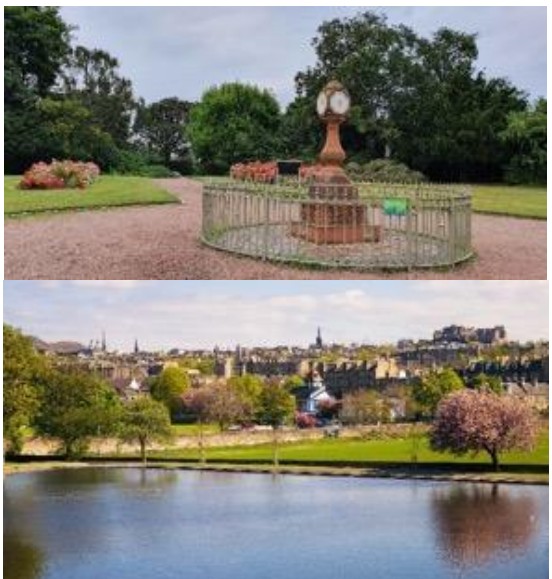
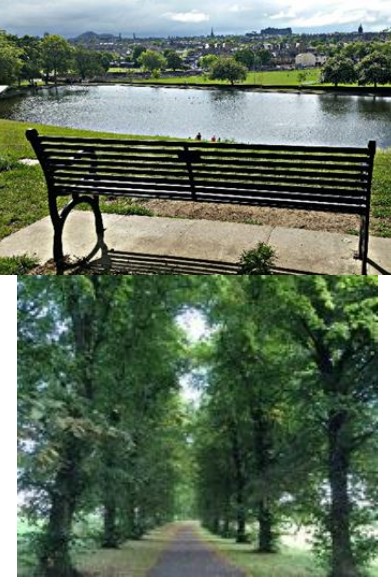

**Figure 13.** Images of Inverleith Park, illustrating the use of zoning to accommodate the variation in individuals' preferences for UGBSs (source: the authors).

These parks also have a variety of vegetation: some areas have trees, hedges, shrubs, and other tall vegetation; some areas have flower beds and lower planting; and some have mown grass and are better suited for play and exercise. Not all UGBS areas are large enough to accommodate this type of zoning, and it is therefore important to consult local communities and ascertain what elements are desired and likely to be utilised to their fullest in each UGBS area. An alternative strategy of providing diversity of character and facility provision across a number of smaller, local parks within relatively close access to each other is another way of accommodating varying needs and preferences in communities and acknowledges the fact that one person's preference may be what deters another.

### 4.3. Future Vision for Urban Green and Blue Spaces

Across the data presented here, which is reinforced by findings from the CEC Thriving Green Spaces visioning workshops with stakeholders (carried out as a separate study with a separate report produced for CEC, which is not reported here), there were five clear topics repeatedly mentioned as positive characteristics of UGBSs, facilitators for visiting these spaces, and related to participants' future vision for UGBSs: 'wilding of our urban green spaces', 'pocket/micro urban green spaces', 'community gardening', 'more trees', and 'green continuous networks'. Hence, we recommend that future research focuses on these five characteristics and investigates to what extent community interventions focusing on these topics can lead to improvements in community engagement with UGBSs and, ultimately, their wellbeing.

However, to improve engagement with lower-income populations and groups living in more deprived urban areas, and to identify what physical features are suitable for and desirable to each community, it is vital to include the local residents in the planning process before any changes happen. A process of co-production for future plans and visions will encourage the residents to take ownership of their local environment and form a stronger place attachment, rather than feeling they are not welcome and that any changes are for others to benefit from, not for them. The Local Government Association recently published a report on co-production with communities to improve social care and mental health and wellbeing. The report highlights the success of co-production and the importance of focusing on place-based, community- and person-centred approaches, to establish successful engagement [49]. It appears that, without a connection to their local UGBS, regeneration and aesthetic improvements alone might not be enough to generate lasting use of, and commitment to, a community's UGBS. This is supported by findings from 'the countryside charity' [50], who reported a significant increase in local green spaces in England between 2012 and 2022. However, they found that, despite an increase in green space designations in deprived areas, a strong correlation persisted between deprivation and use of green spaces. They stated that engagement with community planning may not be a priority for people facing poor housing, low incomes, and barriers to accessing healthcare and other essential services. Furthermore, they concluded that, to address these pressures and create capacity, communities and organisations must work together [50].

Future research should therefore focus on evaluating co-development and community planning processes (i.e., a process where organisations providing public services, business, voluntary groups, and local communities, work together to improve community and individual health), creating the evidence needed to better communicate the benefits of incorporating UGBSs and green features into urban planning, green infrastructure, and development of existing as well as new housing developments.

### 4.4. Limitations

There were major difficulties engaging with the wider public during this project, due to the COVID-19 lockdowns and social distancing rules, meaning that online engagement was the only available method to collect most data. As a result, the Maptionnaire survey was not truly representative of the Edinburgh population, in relation to a number of demographic characteristics and this limits the confidence with which generalisations can

be made over the city as a whole. In addition, the scope of the survey was likely to attract specific sectors and groups already engaged in the issues related to UGBS use, meaning that these responses might be from a biased sample of people already interested in UGBSs and what green and blue infrastructure can offer. The research presented here is therefore only the beginning of a wider process of ongoing engagement that will continue to be required with Edinburgh's residents, workers, and visitors, as the details of new visions are worked out on the ground.

Some population groups were underrepresented in the survey data (individuals from households earning less than GBP37,000, people under 45 years of age, males of all ages, unemployed people, students, and permanently sick or disabled people). From previous research projects involving deprived areas, we know that some communities and individuals are more difficult to engage with than others, and there are a number of challenges in obtaining a truly inclusive response from all different sectors and groups within a large city. When dividing the baseline dataset into subgroups, there were not always sufficient data to infer any conclusive estimates or hypotheses, and larger response numbers would be needed to analyse the data at a neighbourhood level. We recommend that future research should focus on how to engage underrepresented groups to understand better how they might use and benefit from UGBSs.

It is possible that the groups identified as underrepresented via the Maptionnaire survey might not actually be underrepresented in terms of using and benefitting from UGBSs; rather, they may represent a group more difficult to engage in an online survey and less willing to participate in research. Hence, we recommend that future research uses behaviour mapping of specific UGBSs, as well as surveys that directly question park users, to obtain a clearer idea of who visits and what activities they engage in. This would help investigate any true disparity between population subgroups (affluent/deprived, male/female, young/old, etc.) and their use of and engagement with UGBSs.

Since our Maptionnaire data collection did not include questions on respondents' perceived health or quality of life, or how engagements with UGBSs might benefit these, we can only infer the implications of the findings in terms of community health and wellbeing based on the literature [8–13]. This suggests that good quality and quantity of greenery, a feeling of connection with nature, and good overall visual quality act as facilitators for visiting UGBSs and are likely, in turn, to benefit community wellbeing. Communities where these attributes are deficient are, by contrast, less well served in terms of opportunities for wellbeing benefits.

## 5. Conclusions

The overall aim of the research presented here was to investigate the UGBS preferences of diverse populations living in urban areas such as Edinburgh, and the UGBSs they visit. We set out to answer the following questions, as summarised here:

1. Who visits UGBSs in the city, which UGBSs do they visit and/or avoid, how do they get there, and what activities do they engage in?
2. Are the UGBSs visited and/or avoided distinguishable by demographics?
3. What are the characteristics of UGBSs that attracts people and that people dislike?

### 5.1. Visitor Profiles and Patterns

Utilising Maptionnaire, we gathered site-specific data on UGBS usage, enabling the creation of maps illustrating visitor patterns and preferences. Our findings unveiled a myriad of preferences among users, showcasing the complex nature of accommodating diverse needs within constrained geographical areas. We recommend the consideration of ways to accommodate the wide variation in individuals' preferences for UGBSs, including sectioning parks into smaller areas of different characteristics, with different types of vegetation, an array of functions, which are targeted to the needs of diverse groups of visitors.

*5.2. Demographic Influences*

We investigated how demographic factors may shape UGBS utilisation, unearthing disparities in visitation patterns based on age, gender, income, and residential location. Notably, individuals from more affluent areas demonstrated a higher likelihood of visiting UGBSs, while those residing in deprived areas tended to gravitate towards local spaces, indicating limited access beyond their immediate neighbourhoods.

*5.3. Perceived Barriers and Facilitators*

We pinpointed key barriers and facilitators influencing UGBS use, including amenities, accessibility, and environmental quality. Despite nuanced perspectives, overall user satisfaction remained high, emphasising the pivotal role of well-maintained environments in fostering positive experiences.

We found constraints to visiting UGBSs at a distance from home to be stronger for those living in more deprived areas, compared with those whose homes are in les deprived areas. For those living in deprived areas, this may be exacerbated by a poorer quality of the UGBSs close to their homes, meaning that some of these local UGBSs are also places which these respondents avoid. Such barriers need to be addressed by including local communities in the decision-making process and developing future research with a focus on how to engage underrepresented and hard-to-reach groups, ensuring equity in uses of and benefits from UGBSs. Such actions constitute the next phase of CEC's work in taking the Thriving Green Spaces project forward.

**Author Contributions:** Conceptualization, S.B. and C.W.T.; data curation, C.W.-N.; formal analysis, C.W.-N. and Y.W.; funding acquisition, C.W.T.; investigation, C.W.-N. and Y.W.; methodology, C.W.-N., Y.W. and C.W.T.; project administration, C.W.T.; supervision, S.B. and C.W.T.; validation, C.W.-N.; visualization, C.W.-N. and C.W.M.; writing—original draft, C.W.-N.; writing—review and editing, S.B., C.W.M. and C.W.T. All authors have read and agreed to the published version of the manuscript.

**Funding:** Financial support for the project leading to this publication was provided by City of Edinburgh Council (CEC), contract no. C15112. CEC secured funding as lead partner for the TGS project from the Future Parks (FP) Accelerator project grant, in partnership with Edinburgh and Lothians Greenspace Trust, Scottish Wildlife Trust, greenspace scotland, Edinburgh Green Spaces Forum, and the University of Edinburgh. FP is a joint venture between The Heritage Lottery Fund, the National Trust (NT), and the Department for Levelling Up, Housing and Communities [24] to provide funding to preserve the future of the UK's urban parks and green–blue spaces. The United Kingdom Prevention Research Partnership (UKPRP) funded GroundsWell Research Consortium (Grant number: MR/V049704/1) provided additional support for this research. For the purpose of open access, the author has applied a creative commons attribution (CC BY) licence to any author-accepted manuscript version arising.

**Data Availability Statement:** The full data set is not publicly available due to privacy restrictions. However, parts of the data presented in this study are available on request from the corresponding author.

**Acknowledgments:** A series of projects was undertaken in collaboration between the TGS project and students on the Landscape and Wellbeing Masters programme at the University of Edinburgh, as part of their dissertation. Part of the research presented here was undertaken by one of these students, Yiyun Wang, supervised by S.B. and C.W.-N. The contents of the survey were developed by the student, together with her supervisors and staff from CEC. The 'vision' data was collected by CEC, using a Maptionnaire approach, with support and advice from the authors of the paper.

**Conflicts of Interest:** The authors declare no conflicts of interest.

**Appendix A. Map of Edinburgh's Green/Blue Spaces**

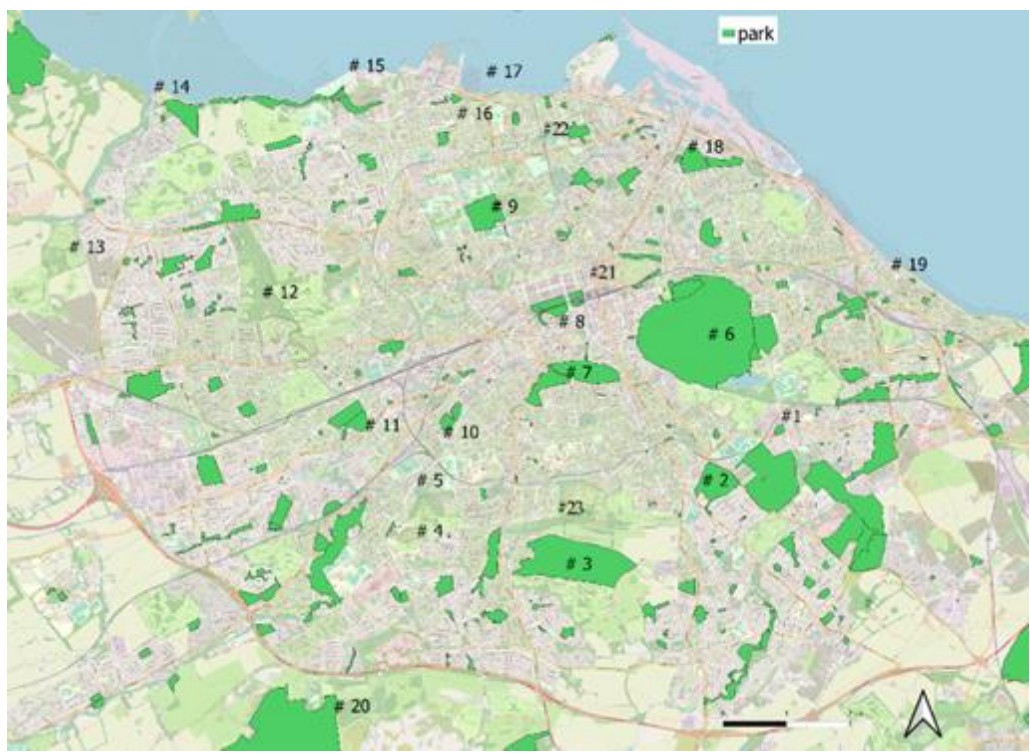

**Figure A1.** Green/blue spaces in Edinburgh. 1: Craigmillar Castle Park; 2: Inch Park; 3: The Braid Hills Golf Course; 4: Craiglockhart Hills; 5: Union Canal; 6: Holyrood Park; 7: The Meadows; 8: Princes Street Gardens; 9: Inverleith Park; 10: Harrison Park; 11: Saughton Park; 12: Corstorphine Hill; 13: Cammo Park; 14: Cramond Seafront; 15: Silverknowes Esplanade; 16: Granton Crescent Park; 17: Wardie Bay; 18: Leith Links; 19: Portobello Beach; 20: Pentland Hills; 21: Calton Hills; 22: Victoria Park; 23: Hermitage of Braid and Blackford Hill.

**Appendix B. Maptionnaire Questionnaire**

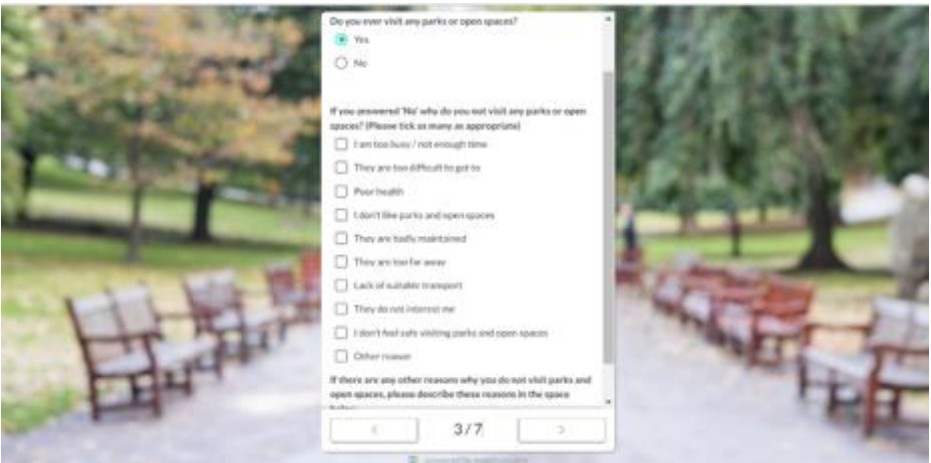

**Figure A2.** Opening question of the TGS Maptionnaire survey: 'Do you ever visit parks or open spaces?'.

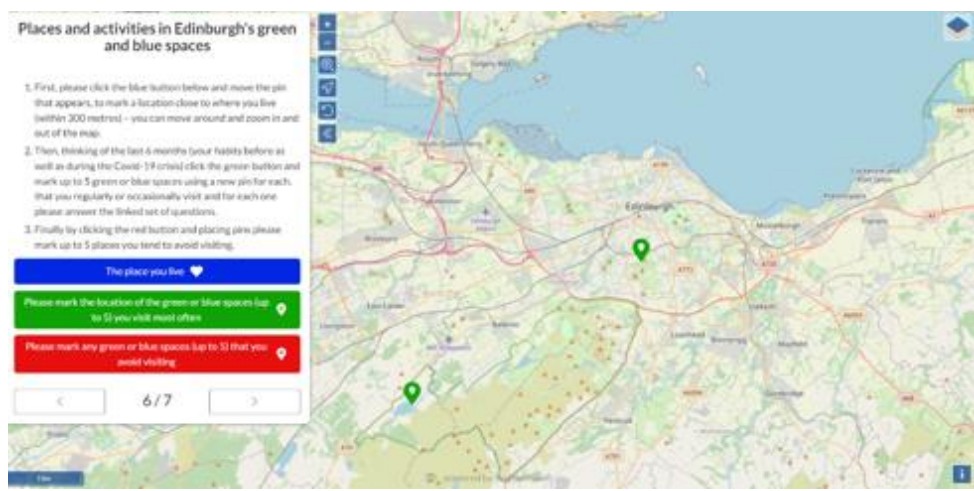

**Figure A3.** For the map-based part of the TGS Maptionnaire survey, the respondents were asked to mark the area where they live, the UGBSs they visit most often, and the UGBSs they avoid visiting.

To investigate further what aspects were regarded as satisfactory when visiting UGBSs, and if certain aspects, or the lack thereof, detracted from the visitor experience, a series of questions were asked that related to each UGBS visited: Q1. Are there toilet facilities in or near the green space? If any, are they usually open (before the lockdown)? Q2. Is the toilet equipped for changing babies or for those with disabilities or special needs? Q3. Does the provision of toilet facilities influence your decision to visit the green space? Q4. Are there cafes in or near the green space? Q5. Does the provision of cafes influence your decision to visit the green space? Q6. Are there shelters from the rain in the green space? Q7. Are there adequate information or sign boards in and near the green space? Q8. Are the information sign boards translated into different languages? Q9. Are there special signs provided for people with disabilities in and near the green space?

**Table A1.** For each UGBS the respondents visited often, they were asked a selection of questions to elicit information regarding visiting patterns and habits.

| Q1. In the Last 6 Months, How Often, on Average, Did You Visit? | Q2. Considering the Impact of the Recent Lockdown. Where You Able to Keep Visiting This Place as Part of Your Daily Exercise Allowance? | Q3. How Do You Travel There? | Q4. How Do You Visit? | Q5. How Long Does You Visit Usually Last? | Q6. What Do You Usually Do There? |
|---|---|---|---|---|---|
| Every day | No | Walk | On my own | <30 min | Walking with/without a dog |
| Several times a week | Yes | Cycle | Whit my partner and friend | 30 min to 1 h | Cycling/ mountain biking |
| Once a week | If yes, did you visit more or less frequently than pre lockdown? | By car | With family | 1 to 2 h | Running/ Jogging |
| Once or twice a month | | By bus | In a group | 2 to 3 h | Playing with children |
| A few times in the last 6 months | | By tram | | 3 to 4 h | Informal games and sports (e.g., frisbee, football, volleyball, etc.) |

Table A1. *Cont.*

| Q1. In the Last 6 Months, How Often, on Average, Did You Visit? | Q2. Considering the Impact of the Recent Lockdown. Where You Able to Keep Visiting This Place as Part of Your Daily Exercise Allowance? | Q3. How Do You Travel There? | Q4. How Do You Visit? | Q5. How Long Does You Visit Usually Last? | Q6. What Do You Usually Do There? |
|---|---|---|---|---|---|
| Not in the last 6 months | | Other | | 4 to 5 h | Through route/ commuting |
| | If other, please specify | | | 5 to 8 h | Eating or drinking |
| | | | | >8 h | Picnic/barbecue |
| | | | | | Visiting an attraction or event (e.g., museum, market, art, concert etc.) |
| | | | | | Art, photography, or similar hobbies |
| | | | | | Sunbathing |
| | | | | | Quiet activities (e.g., reading, meditating) |
| | | | | | Watching wildlife |
| | | | | | Participating in voluntary activities, e.g., Friends of Parks or other groups |
| | | | | | Others |

Table A2. For each of the UGBSs the respondents visited often, they were asked a selection of questions to elicit information regarding the UGBS facilities.

| Questions | Answer Options | | |
|---|---|---|---|
| Q1. Are there toilet facilities in or near the green space? | Yes | No | Don't know |
| If any, are they usually open (before the lockdown)? | Yes | No | - |
| Q2. If there are any toilets, are they equipped for changing babies or for those with disabilities or special needs? | Yes | No | Don't know |
| Q3. Does the provision of toilet facilities influence your decision to visit the green space? | Yes | No | - |
| Q4. Are there cafes in or near the green space? | Yes | No | Don't know |
| Q5. Does the provision of cafes influence your decision to visit the green space? | Yes | No | - |
| Q6. Are there shelters from the rain in the green space? | Yes | No | Don't know |
| Q7. Are there adequate information or sign boards in and near the green space? | Yes | No | Don't know |
| Q8. Are the information sign boards translated into different languages? | Yes | No | Don't know |
| Q9. Are there special signs provided for people with disabilities in and near the green space? | Yes | No | Don't know |

On a Likert scale, participants were asked to rate the following statements: transport links to the site (1: not well connected/bus stops or parking not very close–5: very convenient/bus stop and parking nearby); walking to the site (quality of pavements, slopes, surfaces, street crossings) (1: not very accessible–5: very easy to get to); walking within the site (quality of pavements, slopes, surfaces) (1: not very accessible–5: very easy to move around); facilities (e.g., seating, play equipment, exercise equipment, toilets) (1: poor or inadequate facilities–5: very good and very adequate facilities); maintenance (litter collection, condition of equipment, mowing of grass, etc.) (1: not well maintained–5: very well maintained); trees, shrubs and flowers—quantity and quality (1: not enough trees, shrubs and flowers, poor condition–5: enough trees, shrubs and flowers, in good condition, attractive); safety and security (sense of being threatened, poorly lit with dark places in winter evenings, signs of anti-social behaviour) (1: feels rather unsafe–5: feels very safe); visual quality (attractive overall appearance, views, plants, presence or absence of eyesores) (1: poor visual quality–5: excellent visual quality); connection with nature (1: not connected–5: very connected); interpretation or information provided at the place (1: don't know–5: very good); information to help plan before you visit (e.g., website) (1: don't know–5: very good).

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
