# Peer review of "Your Favourite Park Is Not My Favourite Park: A Participatory Geographic Information System Approach to Improving Urban Green and Blue Spaces—A Case Study in Edinburgh, Scotland"

_land, doi:10.3390/land13030395_

Round 1

Reviewer 1 Report

Comments and Suggestions for Authors

The work presents an exciting approach to the interconnection between green and blue infrastructure and the population's willingness to use it. However, although the authors present the limitations of the work, there needs to be more context focused on factors related to the users' perception of a benefit to their health or improving their quality of life.

In discussing their results, authors could also analyze them from this approach.

Author Response

Dear Reviewer

We would like to thank you for taking the time to review our manuscript. We appreciate your insightful comments and constructive feedback, which have undoubtedly contributed to the improvement of our work.

Below, we address each of your comments and suggestions:

Reviewer 1

Authors Response

Location

The work presents an exciting approach to the interconnection between green and blue infrastructure and the population's willingness to use it. However, although the authors present the limitations of the work, there needs to be more context focused on factors related to the users' perception of a benefit to their health or improving their quality of life.

Thank you for this comment. We have amended the paper’s keywords to reflect this limitation and included the following paragraph under ‘Limitations’ to address this point

Since our Maptionnaire data collection did not include questions on respondents’ perceived health or quality of life, or how engagements with UGBS might benefit these, we can only infer the implications of the findings in terms of community health and wellbeing based on the literature [9-14]. This suggests that good quality and quantity of greenery, a feeling of connection with nature, and a good overall visual quality act as facilitators for visiting UGBS and are likely, in turn, to benefit community wellbeing. Communities where these attributes are deficient are, by contrast, less well served in terms of opportunities for wellbeing benefit.

Line 685

In discussing their results, authors could also analyze them from this approach.

As we have not collected adequate data to make this a legitimate part of our analysis, we have addressed this suggestion as outlined above.

We have carefully considered your suggestions and made the necessary revisions to the manuscript. All changes are all highlighted in yellow in the revised manuscript.

Your comments have undoubtedly enhanced the quality and clarity of our manuscript and we would like to thank you once again for your time and invaluable feedback.

Kind Regards

Reviewer 2 Report

Comments and Suggestions for Authors

This is a useful and interesting study, which makes good use of a combination of GIS and social sciences. With 531 respondents, there is a good basis for making initial statements.

The aim of the TG5 project (of which this study is a part) is to develop the policy (where no infinite amount of money is available) for the protection and enhancement of UGBS (line 76). This research must meet that need. I don't really see that connection yet, although I think it is there.

That's why it surprises me that the emphasis is on the differences between people. The picture that emerges for me from this study is that people think the same about many things and that there are things in which they differ. One solution mentioned is Saugthon Park, where spaces have been created for many different groups. But it is space-consuming, expensive and probably aesthetically unappealing to divide all parks into an (infinite?) number of rooms for many target groups.

Women have different preferences than men, because women feel unsafe in some parks. Wouldn't it be wiser to make all parks safe for women? On average, people from low SES neighborhoods have more difficulty with nearby parks than people in high SES neighborhoods. This is most likely due to the condition of the parks in low SES neighborhoods and says nothing about the preferences of these people. Are there no data on this (line 729: this may be ?exacerbated…., you don't know from your data?).

It makes sense that you separate areas where dogs are allowed to walk off-leash from areas where they must be walked on a leash. It therefore seems useful to me if a set of general positive principles and a number of choice principles are distilled from this study (all still at a theoretical level).

Such a classification also corresponds nicely with the international literature on preferences, which assumes that there is a biological basis for preferences that we share, with a social/economic and personal layer on top of that. See Bourassa, (among others) A paradigm for landscape aesthetics, 1990. I miss the connection with this.

In addition, the introduction talks a lot about health and biodiversity. To what extent has a relationship been established with this in the study?

It also appears that the hypotheses are not all correct. For example: our date revealed no obvious clusters of usage related to age and income (line 504). I would like to know more about why the results do not match the literature. Who is right?

However, a little further down (line 510) says something else: the lack of engagement with UGBS in lower income communities.... Doesn't this contradict line 504?

Author Response

Dear Reviewer

We would like to thank you for taking the time to review our manuscript. We appreciate your insightful comments and constructive feedback, which have undoubtedly contributed to the improvement of our work.
Below, we address each of your comments and suggestions:

Reviewer 2

The aim of the TGS project (of which this study is a part) is to develop the policy (where no infinite amount of money is available) for the protection and enhancement of UGBS (line 76). This research must meet that need. I don't really see that connection yet, although I think it is there.

The aim of TGS project was to contribute to “….improving the city’s natural environment by producing a 30-year strategy and action plan for protecting and enhancing the UGBS… “(line 81). The overall aim of the study we have presented in the manuscript, was to investigate the preferences of residents of Edinburgh and their perceptions of the UGBS they visit. The results and evidence from our study has been used to feed into the larger TGS project, to underpin the new strategy and action plan developed (not presented in the manuscript). The financial aspect of policymaking and UGBS maintenance was not part of the scope for our study and the City of Edinburgh Council’s TGS plan is still out for consultation at the time of writing.

Nonetheless, we have reviewed the paper to ensure we adequately address the reviewer’s comments.

In our discussion (lines 582-585) we suggest that A solution to accommodate the variation in individuals’ preferences for UGBS could be to section parks into smaller defined areas with different types of vegetation, an array of functions, and targeted towards diverse groups of visitors”.

In lines 647-648, we also recommend that, to improve engagement with lower income populations and groups living in more deprived urban areas, “Future research should … focus on evaluating co-development and community planning processes”.

We reiterate these points in the conclusion to ensure we address the reviewers’ comments.

Line 81

Lines 582-585 

Lines 647-648

That's why it surprises me that the emphasis is on the differences between people. The picture that emerges for me from this study is that people think the same about many things and that there are things in which they differ. One solution mentioned is Saugthon Park, where spaces have been created for many different groups. But it is space-consuming, expensive and probably aesthetically unappealing to divide all parks into an (infinite?) number of rooms for many target groups.

In research question 2 we set out specifically to investigate the following: Are the UGBS visited and/or avoided distinguishable by the demographic character of the respondents? And it is therefore a vital part of our study, to investigate the differences between people and how we can ensure that UGBS cater to as many, diverse communities and individuals as possible. In particular, the City of Edinburgh Council wanted to ensure that the city’s UGBS are available, accessible, and desirable to the people who need them the most and are likely to benefit the most from visiting UGBS. This is why there is a focus on differences in people’s preferences in the paper.

We accept the reviewer’s concern that it may not be possible or appropriate to divide all parks into a number of different rooms. In lines 611-614 we note that: An alternative strategy of providing diversity of character and facility provision across a number of smaller, local parks within relatively close access to each other is another way of accommodating varying needs and preferences in communities and acknowledges the fact that one person’s preference may be what deters another.”

A key recommendation (as noted in our response above) is to involve the local communities in the planning and decision-making process, to establish what is needed, and desired, by the local communities.

Lines 611-614

Women have different preferences than men, because women feel unsafe in some parks. Wouldn't it be wiser to make all parks safe for women? On average, people from low SES neighborhoods have more difficulty with nearby parks than people in high SES neighborhoods. This is most likely due to the condition of the parks in low SES neighborhoods and says nothing about the preferences of these people. Are there no data on this (line 729: this may be? exacerbated…., you don't know from your data?).

To make all parks feel safe for women we first need to establish in what areas women do not feel safe and why that is. The preferences of some women are to avoid certain UGBS because they perceive them to be of low or unappealing quality, not necessarily because they are located in a low SES neighbourhood. Thus, we cannot assume that it is the poor condition of parks in deprived areas that is the sole or primary reason women feel unsafe. The literature suggests there are many, complex reasons for women feeling unsafe: for example, a place that is good for wildlife and enjoyed by women who are familiar with the site may be avoided by other women who feel a lack of surveillance, vegetation that restricts views, and/or unfamiliarity with the place make them feel unsafe.  While women’s preferences and concerns are important to address, our data do not allow us to make recommendations beyond noting that there are more places where they feel unsafe than for men. Part of the local community engagement processes that our conclusions recommend would be to investigate why women feel unsafe in some UGBS and what might be done to address this.

It makes sense that you separate areas where dogs are allowed to walk off-leash from areas where they must be walked on a leash. It therefore seems useful to me if a set of general positive principles and a number of choice principles are distilled from this study (all still at a theoretical level).

The idea is to use a co-production model and include local communities in the decision-making process, ensuring equity in use and benefit from UGBS. As an example, some communities might want UGBS where dogs are not allowed at all, and we therefore wanted to avoid generalising. The production of a general set of principles is therefore not within the scope of our study.

We propose including local communities in the decision-making process; avoiding assumptions as to what communities need/want, but rather ask them and listen before (re)designing UGBS.

These actions, however, constitute the next phase of the City of Edinburgh Councils’ work in taking the Thriving Green Spaces project forward. The findings from our study forms the foundation, or baseline, for future refinements and evaluation of the project.

Such a classification also corresponds nicely with the international literature on preferences, which assumes that there is a biological basis for preferences that we share, with a social/economic and personal layer on top of that. See Bourassa, (among others) A paradigm for landscape aesthetics, 1990. I miss the connection with this.

Thank you for this suggestion. We have added a reference to Bourassa’s work in lines 69-72 and to this and other theories in our discussion, lines 566-568.

Lines 69-72

Lines 566-568

In addition, the introduction talks a lot about health and biodiversity. To what extent has a relationship been established with this in the study?

We were not looking to establish a causal pathway between visiting UGBS, biodiversity and health but, rather, we highlight these potential links as reasons for being concerned about who does, or does not, visit UGBS. Investigating such relationships is being addressed in a separate study we are currently working on.

We acknowledge that the paper may have over-emphasised the links to health and have amended the paper’s keywords to reflect this limitation. We have also included the following paragraph under ‘Limitations’ to address this point”

Since our Maptionnaire data collection did not include questions on respondents’ perceived health or quality of life, or how engagements with UGBS might benefit these, we can only infer the implications of the findings in terms of community health and wellbeing based on the literature [9-14]. This suggests that good quality and quantity of greenery, a feeling of connection with nature, and a good overall visual quality act as facilitators for visiting UGBS and are likely, in turn, to benefit community wellbeing. Communities where these attributes are deficient are, by contrast, less well served in terms of opportunities for wellbeing benefit.

Line 684-691

It also appears that the hypotheses are not all correct. For example: our date revealed no obvious clusters of usage related to age and income (line 504). I would like to know more about why the results do not match the literature. Who is right?

Thank you for this comment. We have added the following to address this point:

The literature suggest that, in comparison with young adults, older people may be more concerned with landscape characteristics like legibility, accessibility, and safety, or quality of trails and show less interest in park social participation or vigorous physical activity. Nonetheless, a recent (2018) review also found that older people show quite varied opinions on landscape characteristics, some even contradictory to others. Our findings reflect this variation in preferences which, in our sample at least, may not be significantly different across age groups, while acknowledging that older people, especially those of higher SES, may have the time and resources to travel some distance to find UGBS that suits their particular preferences.

We have reflected this in our discussion on lines 513-522

Line 513-522

However, a little further down (line 510) says something else: the lack of engagement with UGBS in lower income communities.... Doesn't this contradict line 504?

We appreciate the point being raised here, although in line 508 we refer to the lower engagement with UGBS from lower income communities in terms of number of responses. Line 504 refers to the spatial distribution (or clustering) of UGBS engagement, so although both refer to low income communities, there is no contradiction as clustering and absolute number of responses are different metrics.

In line 523, we have added ‘(in terms of UGBS visit numbers)’ to add further clarity here.

Line 508

Line 504

Line 523

We have carefully considered your suggestions and made the necessary revisions to the manuscript. All changes are highlighted in yellow in the revised manuscript. 
Your comments have undoubtedly enhanced the quality and clarity of our manuscript and we would like to thank you once again for your time and invaluable feedback.

Kind Regards

Reviewer 3 Report

Comments and Suggestions for Authors

This paper attempted to use PPGIS, taking Edinburgh, Scotland as a case study, to investigate the preferences of different population subgroups in urban areas and the UGBS they visit. The research is very meaningful and can provide policy recommendations for the construction of UGBS in Edinburgh. However, the text of the paper is not concise enough and the exposition is not standardized enough. Before considering publication, some issues must be addressed. Here are some of my suggestions.

1.      On the first page of the abstract section, it is suggested to supplement the research significance of this study, including theoretical and practical significance.

2.      In line 217, is the gender ratio and white population rate of tourist data in line 217 consistent with the overall data of Edinburgh, and can survey data be used to represent the overall data of Edinburgh? Similarly, in line 245, 54% of the survey responses live in an area from the top quintile (5th quintile, the 20% most influential area) in Scotland. So, is this data representative?

3.      Line 259, does the survey questionnaire indicate how many votes a participant has cast? Is there a weight issue if one person can cast multiple votes?

4.      The conclusion section is somewhat cumbersome, it is recommended to write it in a concise and powerful manner.

5.      Format issues

Line 29, line 479, the sentence does not have a period.

Line 261, 280, 418, and 424 have no empty spaces at the beginning of the paragraph, which is different from the formatting of other paragraphs.

6.      Chart issues

Line 397: The distance between the three sets of text in the legend has not been adjusted to be consistent, which affects the aesthetics

Line 250: The font size in the picture is too large, larger than the font size in the main text. It is also recommended to use pie charts and other graphics for percentage graphics to be more appropriate.

Lines 367 and 463: The text in the picture is default gray, it is recommended to change it to black

Line 580: Suggest image alignment

Line 586: Five images, suggest rearranging the images

Additionally, it is recommended to consider the overall coordination of the entire image when creating images, and to use a uniform font, size, and color as much as possible.

Comments on the Quality of English Language

There are some grammar errors and unclear expressions in the paper. 

Author Response

Dear Reviewer

We would like to thank you for taking the time to review our manuscript. We appreciate your insightful comments and constructive feedback, which have undoubtedly contributed to the improvement of our work.
Below, we address each of your comments and suggestions:

Reviewer 3

On the first page of the abstract section, it is suggested to supplement the research significance of this study, including theoretical and practical significance.

We have now added the following sentence to the abstract to address this point:

PPGIS approaches such as those utilised in this study offer opportunities to address this issue and provide evidence to increase equitable UGBS usage.’

Line 29.

In line 217, is the gender ratio and white population rate of tourist data in line 217 consistent with the overall data of Edinburgh, and can survey data be used to represent the overall data of Edinburgh? Similarly, in line 245, 54% of the survey responses live in an area from the top quintile (5th quintile, the 20% most influential area) in Scotland. So, is this data representative?

Table 1 addresses these questions, showing characteristics of survey respondents compared to sociodemographic data for Edinburgh as a whole.

In section 4.1. we have discussed both these limitations: There were major difficulties engaging with the wider public during this project due to the Covid-19 lockdowns and social distancing rules, meaning that online engagement was the only available method to collect most data. As a result, the Maptionnaire survey was not truly representative of the Edinburgh population in relation to a number of demographic characteristics and this limits the confidence with which generalisations can be made over the city as a whole. In addition, the scope of the survey was likely to attract specific sectors and groups already engaged in the issues related to UGBS use, meaning that these responses might be from a biased sample of people already interested in UGBS and what green and blue infrastructure can offer. The research presented here is therefore only the beginning of a wider process of ongoing engagement that will continue to be re-quired with Edinburgh’s residents, workers and visitors as the details of new visions are worked out on the ground.

Table 1

Section 4.1.

Line 259, does the survey questionnaire indicate how many votes a participant has cast? Is there a weight issue if one person can cast multiple votes?

Line 190 clarifies this: participants were “asked to mark up to five locations of the UGBS they visit most often, and up to five locations of UGBS they avoid visiting”. The locations were not ranked in order of preference. Although each participant could choose more than one location; choosing the same location more than once was not possible.

Line 191

The conclusion section is somewhat cumbersome, it is recommended to write it in a concise and powerful manner.

The conclusion has been edited to address this concern.

Line 692

Line 29, line 479, the sentence does not have a period.

We have now added periods to both sentences.

Line 261, 280, 418, and 424 have no empty spaces at the beginning of the paragraph, which is different from the formatting of other paragraphs.

We have amended each of these lines and each line now has a space at the beginning in alignment with the rest of the manuscript.

Line 397: The distance between the three sets of text in the legend has not been adjusted to be consistent, which affects the aesthetics

We have now amended this to ensure that the three sets of text are equally separated.

Line 250: The font size in the picture is too large, larger than the font size in the main text. It is also recommended to use pie charts and other graphics for percentage graphics to be more appropriate.

The font size has now been changed and we have changed Figure 1 to a pie chart.

Lines 367 and 463: The text in the picture is default gray, it is recommended to change it to black

We have amended both images and the text is now black as requested.

Line 580: Suggest image alignment

Images are now aligned as requested.

Line 586: Five images, suggest rearranging the images

We have rearranged the Figure to only include four images (rather than five). This arrangement of images is now more co-ordinated.

Additionally, it is recommended to consider the overall coordination of the entire image when creating images, and to use a uniform font, size, and color as much as possible.

As mentioned above, we have amended Figures 12 and 13 to improve the the shape, size and co-ordination of the included images.

There are some grammar errors and unclear expressions in the paper. 

We have addressed all of the highlighted grammatical errors and unclear expressions raised here, we have also conducted an additional full text-review to identify and amend any errors which have not been highlighted. These changes have been highlighted yellow in the manuscript.

We have carefully considered your suggestions and made the necessary revisions to the manuscript. All changes are highlighted in yellow in the revised manuscript. 
Your comments have undoubtedly enhanced the quality and clarity of our manuscript and we would like to thank you once again for your time and invaluable feedback.

Kind Regards

Round 2

Reviewer 3 Report

Comments and Suggestions for Authors

The paper can be published

Author Response

Thank you for your comments.